# Isolation, validation, and long-term culture of mouse ear fibroblasts

Katie L. Barnes, Nicole M. Davis, Billy J. Erazo, Katie M. Cataldo, Emmaly H. Bertges and Laura J. Knoll*

## ABSTRACT

Primary fibroblasts are essential cell culture models for studying cellular homeostasis, disease mechanisms, and host–pathogen interactions. Primary cells offer physiological relevance that immortalized cell lines cannot recapitulate. However, existing protocols for isolating and culturing primary fibroblasts lack standardization for long-term applications. Contamination by non-fibroblast cell types, variable media formulations, and poorly monitored passage numbers remain critical challenges for reproducibility. Here, we present a streamlined, cost-optimized protocol for isolating and culturing mouse ear fibroblasts (MEarFs) suitable for extended passages. We demonstrate that enriched media (Advanced DMEM/F-12+15% FBS) consistently promoted adipocyte-like cell contamination in long-term MEarF cultures. In contrast, minimal media (DMEM+10% FBS) proved sufficient for both isolation and long-term cultures, with fibroblast gene expression confirmed through passage 9. Cells displayed characteristic fibroblast morphology, expressed the mesenchymal marker vimentin, and maintained fibroblast-associated transcripts with minimal epithelial and adipocyte marker expression throughout the culture. Bulk RNA sequencing revealed that 17% of mapped genes exhibited differential abundance between passage 3 and passage 9, with enrichment in functional categories related to cell cycle regulation and paracrine signaling. This protocol provides an accessible primary fibroblast cell culture model with broad research applications, emphasizing reproducibility and resource-conscious practices.

KEY WORDS: Primary fibroblast protocol, Cell culture model, Media optimization, Adipocyte-like cell contamination, Transcriptomic profiling

## INTRODUCTION

Fibroblasts are essential structural components of multicellular tissues and serve as an adaptable cell model system for studying cellular homeostasis, disease mechanisms, and host–pathogen infections (Madelaire et al., 2022). Research using fibroblast models relies on either immortalized cell lines or primary cells isolated directly from donors. While immortalized cell lines offer practical advantages, including unlimited proliferative capacity, they frequently fail to recapitulate physiologically relevant cellular processes due to genetic modifications that bypass senescence or their cancerous origins (Shitova et al., 2024). Immortalized fibroblasts are susceptible to misidentification and cross-contamination through prolonged handling across laboratories, contributing to broader concerns about scientific reproducibility. The discovery of HeLa cell contamination in immortalized cell lines, some dating back 30 years, exemplifies these challenges (Hughes et al., 2007; Jäger et al., 2013). Beyond the biological advantages of fibroblasts, practical considerations such as cost, ease of handling, and culture longevity influence model selection. Primary fibroblast protocols that minimize reagent costs and technical complexity while maintaining cellular fidelity would improve accessibility for laboratories and high-throughput studies.

Primary fibroblasts more accurately reflect *in vivo* cellular physiology compared to immortal fibroblasts, as they retain donor-specific genetic and metabolic features. However, primary cells have finite lifespans, requiring researchers to frequently acquire new cells from commercial vendors or isolate them from donors, a process that can be resource-intensive and introduce experimental variability (Chalak et al., 2024). Cell yields and proliferation rates can vary across passages, complicating experimental design and sample size calculations. Both immortalized and primary fibroblast cells undergo molecular and phenotypic adaptations during long-term culture, with the passage number significantly influencing gene expression profiles, metabolic activity, and cellular behavior (Briske-Anderson et al., 1997; Wijerathna-Yapa et al., 2025). With these considerations, primary fibroblasts are preferred for modeling physiological conditions; however, protocols for isolating and culturing primary fibroblasts vary widely in critical steps, such as tissue source, dissociation strategy, and culture media formulations (Table 1). Protocols support short-term culture, with cell viability reported to decline after passage 5. Primary fibroblasts can be successfully isolated from numerous connective tissues, including skin, lung, and heart (Plikus et al., 2021). In rodent models, tail and ear tissues are frequently selected because they are accessible through a noninvasive or nonterminal procedure. Tail-derived fibroblast isolations have been reported to yield inconsistent cell numbers and require extensive enzymatic digestion, increasing cellular stress and experimental costs (Khan and Gasser, 2016). In contrast, ear tissue is amenable to both mechanical and enzymatic digestion strategies, offering flexibility to balance cell yield, culture longevity, and experimental cost (Uguten et al., 2024). Ear tissue can be collected terminally or through nonterminal ear punches, enabling parallel experiment design with *in vitro* and *in vivo* systems (Edelman and Redente, 2018).

A critical, but underappreciated, challenge in primary fibroblast isolation is contamination by nonfibroblast cell types (Challener, 2024). Mouse ear tissue contains heterogeneous cell populations, including fibro-adipogenic progenitors, cells that can differentiate into both fibroblasts and adipocytes (Joe et al., 2010; Marangoni et al., 2020). While enriched media formulations may accelerate initial fibroblast outgrowth, they can also promote adipogenic differentiation, leading to mixed-population cultures (Roxburgh et al., 2016). Adipocyte-like contaminants can proliferate and ultimately outcompete fibroblasts in enriched media, compromising culture identity and experimental interpretation. Furthermore, lipid

Department of Medical Microbiology and Immunology, University of Wisconsin-Madison School of Medicine and Public Health, Madison, WI 53706, USA.

*Author for correspondence (ljknoll@wisc.edu)

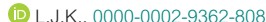 L.J.K., 0000-0002-9362-8088

**Table 1. Comparison of five fibroblast isolation protocols**

| Protocol reference | Base media | Serum | Cell isolation | Validation | Tissue and species | Notes |
|---|---|---|---|---|---|---|
| Ningsih et al., 2024 | DMEM – low glucose, pyruvate (Thermo Fisher #11885084) | 10% FBS (Sigma-Aldrich #F2442) | Mechanical | Flow cytometry of CD73 and CD90, or CD105, and immunofluorescence imaging of vimentin | Human skin (normal and pathological conditions) | Identifies 'adipocyte-like cells'; suggests the need for more antibiotic/bacterial/ mycotic testing for long-term culture |
| Edelman and Redente, 2018 | DMEM (not listed) | 10% FBS (not listed) | Enzymatic (Collagenase A) or mechanical | Phalloidin stain | Mouse lung | Adapted for ear punch or tail snips |
| Seluanov et al., 2010 | DMEM/F-12 and EMEM (not listed) | 15% FBS (not listed) | Mechanical and enzymatic (Liberase) | Morphology | Rodent (>20 wild species tested) skin and lung | DMEM/F-12 for isolation and initial growth period; EMEM for fibroblast selection media |
| Khan and Gasser, 2016 | RPMI 1640 (not listed) | 10% FBS (not listed) | Enzymatic (Collagenase D and Pronase) | Vimentin labeling and imaging | Mouse ear and tail | Validated cell isolation from tissue stored at room temperature up to 10 days; maximum of six passages to signs of senescence |
| Bravo et al., 2021 | DMEM (Corning #15-013-CV) | 10–20% FBS (Sigma-Aldrich #F0926-500ML) | Mechanical | Morphology, flow cytometry of THY1.2 or PDGFRα | Mouse ear | Suggests 20% FBS for long-term culture; validation at passage 3; maximum of ten passages |
| This paper | DMEM (Gibco #11960-051) | 10% FBS (Corning #35-010-CV) | Mechanical and enzymatic (Liberase) | Morphology, vimentin staining, and RNA sequencing | Mouse ear | Validated up to passage 9; no adipocyte or myocyte contamination reported in minimal media, which is nutrient sufficient for fibroblast isolation; published transcriptome of passage 3 and passage 9 cells |

Base media, serum, cell-isolation mechanism, method of validation, and tissue type used for isolation are among the major differences between the published protocols.

metabolism in contaminating cell populations may exert paracrine-like effects on neighboring cells, confounding experimental results and reducing the physiological relevance of the model system (Else, 2019). Despite these concerns, published protocols for primary fibroblast isolation rarely address nonfibroblast contamination or its prevention.

Media composition is another significant variable in primary fibroblast culture development. Primary cells often rely on selective media formulations to support survival and growth, yet these conditions can also influence cell physiology and metabolism (Wijerathna-Yapa et al., 2025). Published primary fibroblast isolation protocols recommend a wide range of base media formulations, from minimal (DMEM+10% FBS) to highly enriched (Advanced DMEM/F-12+15% FBS), with variable passage longevity and limited characterization of contaminating cell types (Table 1). Enriched media may enhance initial fibroblast growth, but long-term culture in enriched media remains poorly characterized. Minimal media formulations offer a simpler, more cost-optimized approach while selectively supporting fibroblast survival over contaminating cell types.

Given the need for reproducible, cost-optimized primary fibroblast models suitable for extended fibroblast culture, we developed and validated a streamlined protocol for isolating

mouse ear fibroblasts (MEarFs). Our approach addresses three critical gaps identified in existing protocols (Table 1): minimizing nonfibroblast contamination during long-term culture, identification of optimal media conditions to balance cost with culture purity, and documentation of passage-dependent transcriptomic changes relevant to experimental design with the model. The protocol we outline here provides an accessible, cost-optimized approach to primary fibroblast culture with applications in cellular homeostasis, disease modeling, and host–pathogen interactions.

## RESULTS
### Streamlined isolation of primary fibroblasts from mouse ear tissue

To address the need for a reproducible and longer-lived primary fibroblast model, we established and validated a streamlined protocol for isolating MEarFs as a suitable cell model. Our protocol integrates mechanical and enzymatic dissociation steps adapted from previously published protocols (Table 1). We collected tissue from both ears of a single mouse (Fig. 1A). Using a blade, we mechanically disrupted the tissue by breaking it down into smaller fragments (Fig. 1B). We further digested the tissue fragments using the enzyme Liberase for 90–120 min (Fig. S1). Liberase was chosen for enzymatic tissue disruption because of the documented success of tissue degradation

## A. Tissue collection

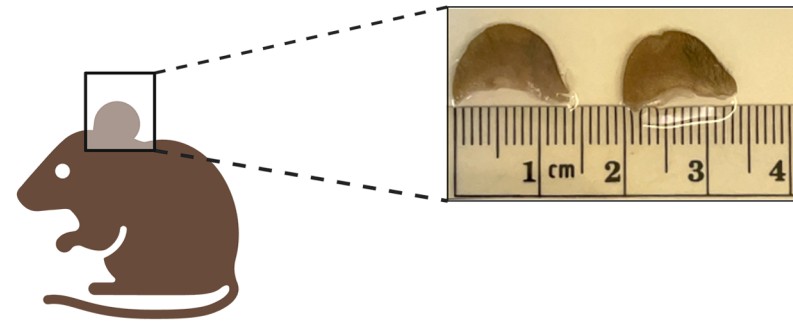

## B. Tissue processing

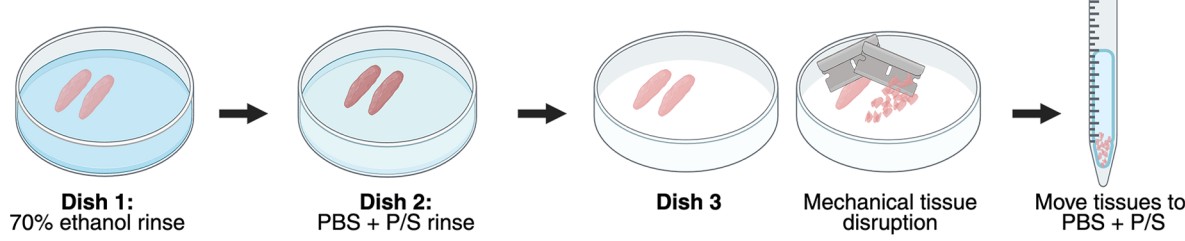

| Dish 1:<br>70% ethanol rinse | Dish 2:<br>PBS + P/S rinse | Dish 3 | Mechanical tissue<br>disruption | Move tissues to<br>PBS + P/S |

## C. Cell isolation

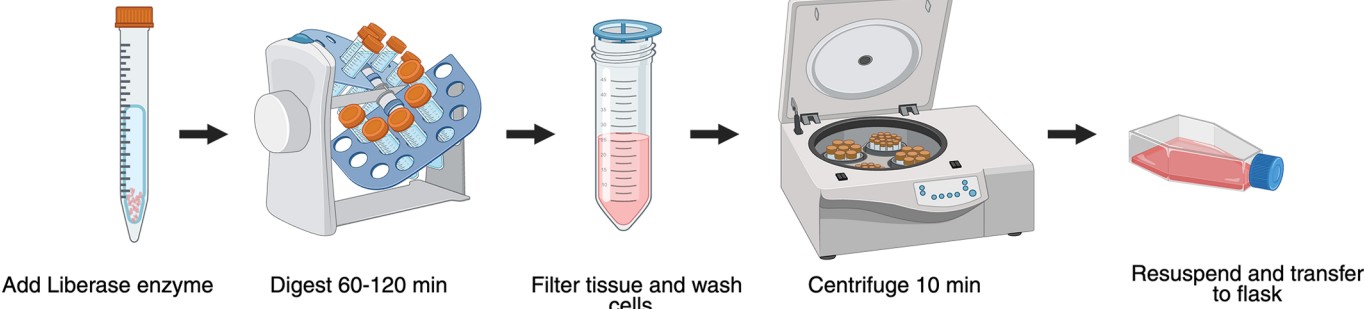

| Add Liberase enzyme | Digest 60-120 min | Filter tissue and wash<br>cells | Centrifuge 10 min | Resuspend and transfer<br>to flask |

**Fig. 1. Workflow for mouse ear tissue collection and processing for primary fibroblast culture.** (A) Ear tissue was collected from 16-week-old female C57BL/6J mice following humane euthanasia in accordance with institutional guidelines. (B) The tissue was washed in 70% ethanol, rinsed in phosphate-buffered saline (PBS) supplemented with 1% penicillin–streptomycin (P/S), and then mechanically disrupted with razor blades. The minced tissue was transferred to PBS+1% P/S containing the Liberase enzyme and digested for 90–120 min at 37°C with rotation. (C) After digestion, the remaining tissue fragments were removed by filtration, and the cell pellet was washed to remove residual enzyme in minimal media (DMEM+10% FBS). Cells were seeded into a T25 cell culture flask and left undisturbed for 3 days to aid cell adhesion. Created in BioRender by Knoll, L. J. (2026). https://BioRender.com/62oxy8p. This figure was sublicensed under CC-BY 4.0 terms.

as a mix of Collagenases I and II and proteases (Brandhorst et al., 2010). Following digestion, we removed tissue fragments by filtration to reduce co-isolation of nonfibroblast cell types. We then washed the cells, resuspended them in DMEM supplemented with 10% FBS, and seeded them into a T25 flask for culture (Fig. 1C). The flask was left undisturbed for 3 days to aid cell adhesion.

### Minimal media reduces adipocyte-like contamination in MEarF culture

Media composition influences cellular metabolism and behavior (Lee et al., 2018; Vis et al., 2020; Cataldo et al., 2026). Enriched

media may support the initial outgrowth of fibroblasts; however, excessive nutrients can also promote the survival and expansion of less-resilient cell populations. For example, fibro-adipogenic progenitors are skeletal muscle cells that can be isolated alongside fibroblasts (Joe et al., 2010; El-Hattab et al., 2020). With this in mind, we examined whether MEarFs could be maintained long term in minimal media (DMEM+10% FBS) to reduce exposure to multiple media formulations and limit culture-media-associated variability. By passage 3, MEarFs cultured in enriched media (Advanced DMEM/F-12+15% FBS) accumulated adipocyte-like cells within the monolayer, as confirmed by boron–dipyrromethene

(BODIPY) staining of neutral lipids (Fig. 2A, green). In MEarF cultures grown in minimal media, adipocyte-like cells are undetectable (Fig. 2B). Quantification between two independent experiments demonstrates a significant reduction in the number of BODIPY-positive cells when grown in minimal media compared to enriched media (Fig. 2C). Cells with an accumulation of lipid-like structures were found in MEarF cultures in enriched media, regardless of the donor's genotype (Fig. S2A). Cells imaged without BODIPY stain demonstrated a lack of specific fluorescence, confirming the BODIPY stain co-localized with adipocyte-looking cells (Fig. S2B).

To determine whether adipocyte-like cells could be eliminated through trypsin exposure and passaging, we cultured MEarFs isolated from the same donor, in three conditions: enriched media from isolation to passage 11 (Fig. 2D); enriched media for five passages followed by minimal media for another five passages

(Fig. 2E); and minimal media from isolation to passage 13 (Fig. 2F). Adipocyte-like cells overgrew cultures maintained in enriched media by passage 11, with fibroblasts largely outcompeted for space. Cultures initiated with enriched media and transitioned to minimal media displayed a reduced but persistent population of adipocyte-like cells. MEarFs cultured in minimal media for at least two passages did not exhibit an accumulation of adipocyte-like cells (Fig. 2B; Fig. S4). Adipocyte-like cells did not grow to a noticeable amount through passage 13, as determined by morphology under light microscopy. Taken together, MEarFs grown in minimal media have reduced adipocyte-like cell contamination.

## Validation of fibroblast identity in MEarF cultures

Following the successful reduction of adipocyte-like contaminants, we validated fibroblast identity in MEarF cultures using immunofluorescent, morphological, and transcriptomic analyses.

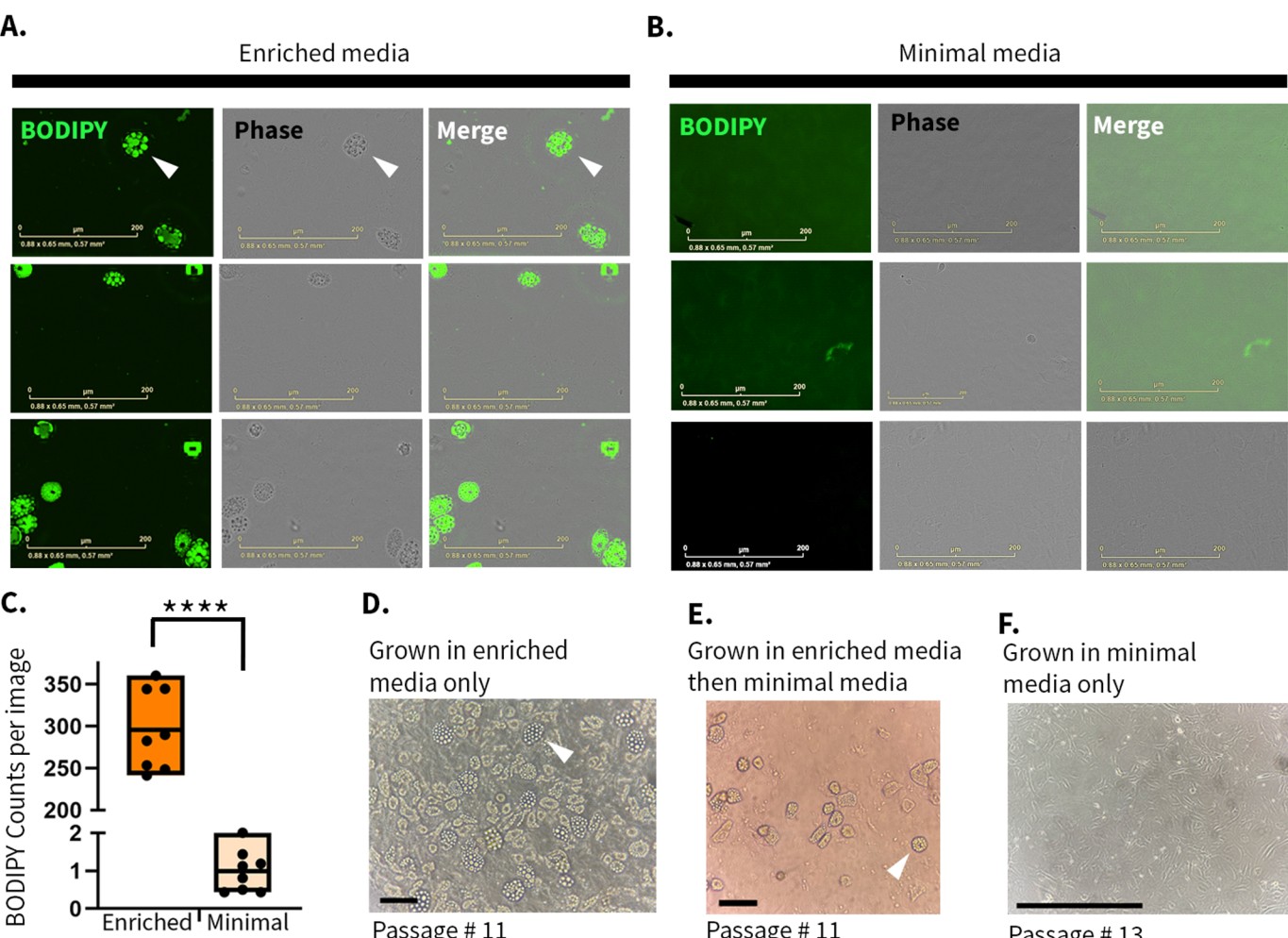

**Fig. 2. Minimal media reduces adipocyte-like contamination in MEarF cultures.** (A,B) Passage 3 MEarFs were stained with the neutral lipid dye, BODIPY, to confirm the lipid identity of cell contaminants. Cells were cultured in enriched media (Advanced DMEM/F-12+15% FBS; A) or minimal media (DMEM+10% FBS; B) from isolation through passage 3 (P3). Images were acquired using GFP and phase-contrast channels on an Incucyte live-cell imaging system with a 20× objective, scale bars: 200 µm, n=3. Localized green fluorescence indicates BODIPY-positive adipocyte-like cells. (C) Quantification by Incucyte Sartorius software for BODIPY counts per image demonstrates the significance (****P<0.0001) of lipid-like cells in enriched compared to minimal media conditions. (D–F) MEarFs isolated from the same donor, maintained under three media conditions. MEarFs grown in enriched media (D) from isolation to passage 11 accumulated an abundance of lipid-like cells (white arrowhead). MEarFs grown in enriched media from isolation to passage 5 and then minimal media from passages 5 to 11 (E), generating a moderate number of lipid-like cells (white arrowhead). When grown in minimal media from isolation to passage 13 (F), MEarF cultures have an undetectable amount of lipid-like cells. Images were captured from one flask for each condition through the eyepiece of a 10× (D,E) and 20× (F) objective light microscope on a phone camera, scale bars: 200 µm, n=2. Cultures maintained exclusively in minimal media showed no detectable adipocyte-like cells throughout extended culture in at least four separate isolations.

Immunofluorescent staining for vimentin, a mesenchymal marker, confirmed fibroblast lineage within the culture of MEarFs, compared to the positive control of human foreskin fibroblasts (HFFs) and the negative control of epithelial Madin-Darby canine kidney (MDCK) cells (Fig. 3A). MEarF cultures further displayed characteristic fibroblast morphologies: adherent, spindle-shaped cells with distinct nuclei and cytoplasm appearing 10–20 days post-isolation (Fig. 3B). Secondary-only controls (Fig. S3) showed a pattern of background fluorescence that differentially localized compared to the Vim primary antibody-positive cells.

Fibroblast identity was confirmed by comparing RNA sequencing of passage 3 and passage 9 MEarFs to published RNA-sequencing datasets of adipocyte-like and epithelial–mesenchymal transition cells as a reference for key genes (Fig. 3C). Raw mapped reads from 3T3-L1 adipocyte–fibroblasts were downloaded from the R packages curatedAdipoRNA (R package version 1.26.0, https://bioconductor.org/packages/curatedAdipoRNA) and ExperimentHub with the emtdata library (R package version 1.18.0, https://bioconductor.org/packages/emtdata). DESeq2 analysis of all six cell populations was done in tandem to ensure minimal variation in analysis. Normalized counts of vimentin (*Vim*) and forkhead box C2 (*Foxc2*) were higher in MEarF transcriptomes at both passages 3 and 9 compared with the 3T3-L1 and human mammary epithelial (HMLE) reference sequences. Epithelial cadherin (*Cdh1*) and epithelial cellular adhesion molecule (*Epcam*) have a markedly lower normalized count value in MEarF cells compared to both populations of HMLE cells. Adipocyte gene markers such as fatty acid binding protein (*Fabp4*) and peroxisome proliferator-activated receptor-gamma (*Pparg*) were detected exclusively in adipocyte-like 3T3-L1 cells. Leptin (*Lep*) and adiponectin (*Adipoq*) are other key adipocyte markers that are absent in passage 3 or passage 9 MEarFs (Table 2). Housekeeping gene (Nazet et al., 2019) counts varied between MEarFs at passages 3 and 9. Compared to 3T3-L1 adipocyte-like fibroblasts, and HMLE epithelial-mesenchymal reference cells, MEarFs maintained a relatively moderate abundance of glyceraldehyde-3-phosphate dehydrogenase (*Gapdh*) and ribosomal protein lateral stalk subunit P0 (*Rplp0*).

### Transcriptomic adaptation of MEarFs during extended culture

Despite consistent seeding densities, we observed substantial variability in total cell numbers throughout passages, suggesting continued adaptation of the cells to *in vitro* conditions (Jia et al., 2022). Cell counts from a T25 culture flask rose from 430,000 at passage 3 to 830,000 by passage 9 (Table 3), prompting our investigation into transcriptional changes associated with long-term culture.

We performed RNA sequencing on the population of MEarFs at passages 3 and 9. Our bioinformatic analysis began with principal component analysis (PCA), separating samples by passage number and clustering with biological replicates (Fig. 4A). Differential expression analysis identified significant transcriptional shifts, with 17% of mapped genes exhibiting differential abundance between passages (Fig. 4B). Hierarchical clustering of the 500 most differentially abundant genes further segregated samples by passage number (Fig. 4C). Gene ontology (GO) enrichment analysis highlighted functional categories associated with chemotaxis, epithelial cell proliferation, and leukocyte migration (Fig. 4D). These results highlight the importance of tracking the passage number of primary cells for experimental reproducibility.

### DISCUSSION

We developed a simplified, cost-optimized, and reproducible approach for isolating and culturing primary fibroblasts long term.

We demonstrate that minimal media (DMEM+10% FBS) is sufficient for both the isolation and long-term culture of MEarFs, offering a practical alternative to enriched media formulations. Minimal growth conditions selectively support fibroblasts and provide a stable culture environment, reducing experimental variability and improving reproducibility across studies. In contrast, enriched media (Advanced DMEM/F-12+15% FBS) consistently promoted the expansion of adipocyte-like cells, indicating that the MEarF culture is highly sensitive to media conditions.

We validated the use of two mouse ears per digestion as a reliable source of MEarFs. Consistent with previous isolation protocols (Edelman and Redente, 2018), we successfully isolated MEarFs using just two ear hole punches without filtering out the tissue, as we suggest doing with larger tissue inputs. MEarFs from two hole punches did take a total of 34 days to reach confluence in a T25 flask, compared to both ears from a 3-week-old donor (7 days to confluence) or a 12-week-old donor (12 days to confluence) (Fig. S4). Our protocol may be adapted by growing the initial culture from two to four hole punches in a 6 cm Petri dish to aid confluency before transitioning to a culture flask. We did not characterize the long-term proliferative capacity or transcriptomic adaptations of the cells isolated from a smaller tissue input. Nonlethal tissue collection offers the advantage of establishing parallel *in vitro* models from animals undergoing *in vivo* experimentation but will require further optimization.

Adipocyte-like cells observed in MEarF cultures were first believed to be co-isolated cells during tissue processing, consistent with prior reports of adipocyte-like cell contamination in primary human fibroblast cultures (Ningsih et al., 2014). Media formulation was not the first variable we adjusted to minimize the adipocyte-like contamination. Notably, the mouse primary fibroblast isolation protocols with the most enriched media did not report adipocyte-like contamination (Seluanov et al., 2010). However, in that study, researchers isolated fibroblasts from wild-caught mice, which exhibit more efficient lipid metabolism than inbred laboratory mice (Martin et al., 2010), potentially influencing the abundance or differentiation capacity of adipogenic precursor cells, depending on the background of the donor mouse. Additionally, other primary fibroblast isolation protocols (Table 1) characterize cells only at early passages, whereas we observed the cells across multiple passages, so contaminating cells have time to proliferate and expand during our observations.

Donor age was hypothesized to contribute to the appearance of differentiating adipocyte cells, so we isolated MEarFs from 3-week-old mice and compared them with those from 12-week-old adult mice (Berry et al., 2013; Bruder and Fromme, 2022). When grown in enriched media, MEarFs from young donors still produced adipocyte-like cells, suggesting that donor age alone is insufficient to prevent co-isolated cell contaminants. In contrast, MEarFs isolated in minimal media, from 3-week, 6-week, and 12-week-old mouse donors, reached confluence in minimal media without adipocyte contamination (Fig. S4). This suggests that media composition, not the age of the donor, is likely responsible for the proliferation of adipocyte-like contaminants. We observed that the only donor-age-related difference was that young donors reached confluence post-isolation sooner than the older donors; however, beyond passage 0, donor age did not significantly influence confluency rate.

Although the origin of the adipocyte-like cells in our isolations remains unclear, their morphology is consistent with co-isolated fibro-adipocyte progenitor (FAP) cells that differentiate toward adipocytic phenotypes under enriched and stressful culture

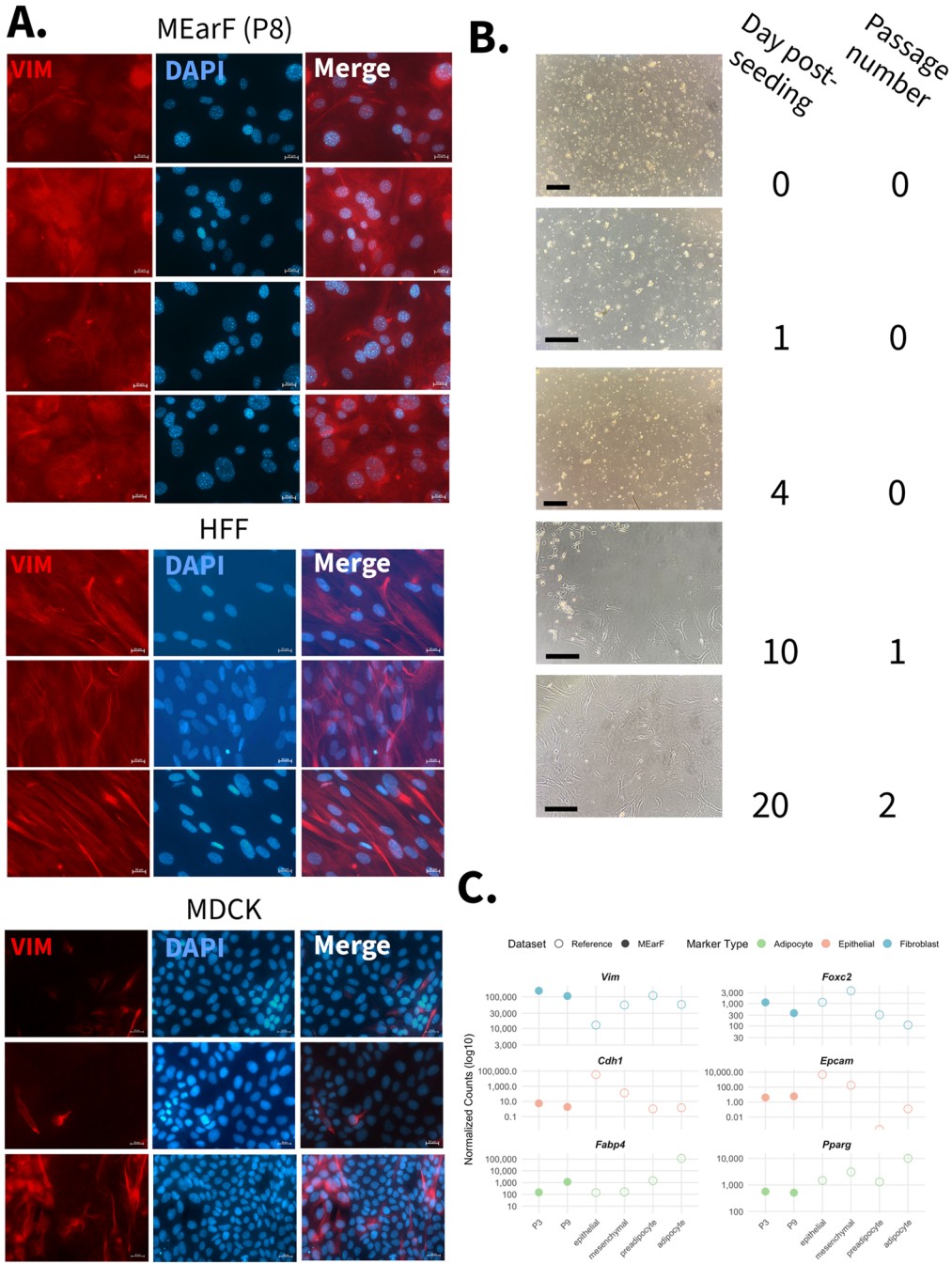

**Fig. 3. Confirmation of fibroblast identity in mouse ear fibroblast (MEarF) cultures.** (A) Immunofluorescence staining for vimentin (VIM, red), a fibroblast-specific intermediate filament protein, in passage 8 MEarFs compared to reference cell lines. Human foreskin fibroblasts (HFFs) serve as a positive control for fibroblast identity, while Madin-Darby canine kidney epithelial cells (MDCK) serve as a negative control. Nuclei were counterstained with DAPI (blue). Images were acquired on a Zeiss Axioplan III fluorescence microscope at 40× magnification, scale bars: 20 µm. (B) Morphological progression of the MEarF monolayer across 20 days post-isolation. Cells transitioned from initial suspension to adherent monolayers displaying characteristic fibroblast morphology: elongated, spindle-shaped cells with distinct nuclei and cytoplasm by day 10. Images shown are taken at 0, 1, 4, 10, and 20 (passages 0, 1, and 2). Images are taken through the eyepiece of a 10× objective light microscope with a phone. Scale bars: 200 µm. (C) Comparative transcriptomic validation of MEarF fibroblast identity. MEarFs (closed dots) show expression patterns consistent with fibroblast identity: high expression of fibroblast markers in blue (*Vim*, *Foxc2*), minimal expression of epithelial markers in orange (*Cdh1*, *Epcam*), and intermediate expression of adipocyte markers in green (*Fabp4*, *Pparg*) relative to reference datasets (open dots). Expression of cell-type-specific markers in MEarFs [passages 3 (P3) and 9 (P9)] are compared to reference datasets of fibro-adipogenic cells and epithelial–mesenchymal transition cells: 3T3-L1 cells (pre-adipocyte versus differentiated adipocytes) and human mammary epithelial (HMLE) cells (epithelial versus mesenchymal). MEarF transcriptomic data were generated in-house as described in Materials and Methods (*n*=4). Reference datasets were obtained from Bioconductor: 3T3-L1 data from the curatedAdipoRNA package (*n*=22, pre-adipocyte, *n*=25 adipocyte-like) and HMLE data from the emtdata package via ExperimentHub (*n*=3). Raw counts from all datasets were normalized using DESeq2 to enable direct comparison. Values represent log-transformed normalized counts.

**Table 2. Direct comparison of normalized counts of differentially expressed transcripts by DESeq2 transcriptomic analysis of passage 3 and passage 9 mouse ear fibroblasts (MEarFs), pre-adipocyte and adipocyte fibroblasts (3T3-L1s), and immortalized epithelial and mesenchymal differentiated cells (HMLEs)**

| | Top genes in GO term Nuclear Division | | | | | | | | |
| --- | --- | --- | --- | --- | --- | --- | --- | --- | --- |
| | Cdc20 | Cyp26b1 | Ereg | Fgfr2 | Kifc1 | Mastl | Npr2 | Prickle1 | Spire1 |
| Passage 3 MEarF | 4278 | 6325 | 2341 | 4730 | 1314 | 1170 | 4502 | 733 | 1589 |
| Passage 9 MEarF | 1421 | 1270 | 860 | 1478 | 477 | 423 | 2197 | 2473 | 479 |
| Pre-adipocyte 3T3-L1 | 840 | 688 | 396 | 2179 | 388 | 271 | 2840 | 1425 | 621 |
| Adipocyte-like 3T3-L1 | 220 | 251 | 88 | 3185 | 99 | 125 | 1652 | 1157 | 908 |
| Epithelial HMLE | 19,543 | 495 | 0 | 1202 | 7242 | 1791 | 980 | 301 | 2701 |
| Mesenchymal HMLE | 12,519 | 172 | 0 | 35 | 7284 | 2120 | 2224 | 106 | 2780 |

| | Top genes in GO term Regulation of Epithelial Cell Proliferation | | | | | | | | |
| --- | --- | --- | --- | --- | --- | --- | --- | --- | --- |
| | Cxcl12 | Ccnd1 | Apln | Jag1 | Twist2 | Fgfr2 | Zfp703 | Cyp7b1 | Fgf2 |
| Passage 3 MEarF | 33,048 | 15,070 | 412 | 2489 | 949 | 6817 | 9302 | 366 | 6817 |
| Passage 9 MEarF | 138,117 | 2539 | 4632 | 390 | 6411 | 1953 | 2315 | 2039 | 1953 |
| Pre-adipocyte 3T3-L1 | 55,976 | 10,274 | 493 | 2389 | 2341 | 37 | 2436 | 1337 | 37 |
| Adipocyte-like 3T3-L1 | 15,956 | 2127 | 1953 | 2948 | 839 | 118 | 3874 | 362 | 118 |
| Epithelial HMLE | 3 | 17,251 | 2131 | 0 | 1263 | 1032 | 0 | 36 | 1032 |
| Mesenchymal HMLE | 49 | 15,579 | 174 | 0 | 1173 | 3850 | 0 | 5 | 38,450 |

| | Quiescence | | | | | | Senescence | | | | |
| --- | --- | --- | --- | --- | --- | --- | --- | --- | --- | --- | --- |
| | Pi16 | Dpt | Cd34 | Mfap4 | Ly6a | Fbln1 | Ccnd1 | Cdkn2b | E2f1 | Hdac1 | Jun |
| Passage 3 MEarF | 31 | 309 | 756 | 258 | 12,610 | 3441 | 15,070 | 707 | 1255 | 5481 | 6474 |
| Passage 9 MEarF | 316 | 1819 | 799 | 743 | 24,828 | 6886 | 2539 | 530 | 1179 | 5044 | 7629 |
| Pre-adipocyte 3T3-Ll1 | 188 | 3121 | 25 | 469 | 3040 | 3509 | 10,274 | 2692 | 1203 | 1419 | 2210 |
| Adipocyte-like 3T3-L1 | 59 | 2900 | 14 | 179 | 790 | 704 | 2127 | 1059 | 773 | 811 | 1559 |
| Epithelial HMLE | 7 | 1 | 38 | 19 | 0 | 6920 | 17,251 | 1640 | 0 | 10,116 | 9510 |
| Mesenchymal HMLE | 14 | 7 | 85 | 107 | 0 | 9451 | 15,579 | 1960 | 0 | 9574 | 10,623 |

| | Housekeeping | | Adipocyte | | | Epithelial | | Fibroblast | | |
| --- | --- | --- | --- | --- | --- | --- | --- | --- | --- | --- |
| | Gapdh | Rplp0 | Adipoq | Fabp4 | Lep | Cdh1 | Epcam | S100a4 | Vim | Foxc2 |
| Passage 3 MEarF | 81,209 | 78,872 | 3 | 148 | 0 | 0 | 7 | 41,318 | 154,564 | 1122 |
| Passage 9 MEarF | 117,105 | 93,850 | 0 | 1165 | 0 | 3 | 1 | 20,679 | 105,646 | 374 |
| Pre-adipocyte 3T3-L1 | 3098 | 47,326 | 43 | 1459 | 0 | 1328 | 0 | 17,237 | 109,117 | 316 |
| Adipocyte-like 3T3-L1 | 2689 | 26,137 | 73,457 | 115,437 | 243 | 1229 | 0 | 6621 | 56,773 | 110 |
| Epithelial HMLE | 208,706 | 50,667 | 0 | 130 | 0 | 19,280 | 5906 | 410 | 11,796 | 1055 |
| Mesenchymal HMLE | 152,187 | 37,612 | 0 | 213 | 0 | 12,242 | 4947 | 295 | 55,863 | 3294 |

Genes in this table are grouped by functional category and signature marker for the listed cellular function. Top genes influencing GO categories Regulation of Epithelial Cell Proliferation and Nuclear Division are represented, as well as the values for housekeeping, adipocyte-like, and fibroblast-like normalized counts shown in Fig. 3C. Normalized count values were calculated using DESeq2 and are rounded to the nearest whole number.

conditions (Marangoni et al., 2020). FAP cells are heterogeneous and serve as key regulators of muscle regeneration and stem cell differentiation (Molina et al., 2021). Importantly, the presence of FAP cells is an indicator of culture stress and dysregulation as markers of wound healing (Loomis and Smith, 2023). We hypothesize that FAP cells proliferated in our cultures due to enriched media conditions and prolonged culture stress. This suggests that our protocol could be adapted to intentionally establish FAP cell models, providing a platform for studying *in vitro* fibro-adipogenic differentiation.

Reducing adipocyte-like contamination was essential for two reasons. First, adipocyte-like cells preferentially proliferated in enriched media and ultimately outcompeted fibroblasts, compromising culture identity. Second, lipid metabolism exerts broad paracrine signals on surrounding cells, while standard cell culture conditions already impose an artificial nutrient environment that perturbs cellular metabolism from physiological conditions (Else, 2019). The combination of enriched media and lipid-rich contaminant populations exhibits compounding derivations from physiological conditions, undermining the translational relevance of fibroblast-based models.

Myocytes are noted contaminants of published fibroblast isolation protocols, and we did not see such contamination in our cells grown in minimal media. The absence of myocyte contamination in our MEarFs may be due to a few reasons. First, we remove tissue chunks immediately after mechanical and enzymatic tissue digestion. Second, myocytes take time to divide and differentiate, so our care of fibroblasts may be selective against the growth of these cells (Uezumi et al., 2016). Finally, we confirmed the absence of myoblasts through the absence of marker genes, *Myod* and *Myf5*, in MEarF transcriptomics.

When grown in optimal media conditions, MEarFs exhibited adaptability to cell culture over time within a finite window of viability. Variability in total cell counts across passages (Table 3) prompted us to investigate transcriptional differences between early- and late-passage cells. Bulk RNA sequencing of MEarFs revealed substantial transcriptomic shifts between passage 3 and passage 9. PCA demonstrated clear clustering of samples by passage number, with larger variation across passage 3 than passage 9 samples (Fig. 4A). Differential expression analysis identified significant transcriptional changes, with 17% of mapped genes exhibiting differential abundance between passages 3 and 9

**Table 3. An example of MEarF passages over the course of 3 months**

| Date | Passage number | Days since last passage | Total cells | Delta cells (total cells of this split − seeding density of last passage) | Seeding density (number of cells) | Cells/day |
|---|---|---|---|---|---|---|
| 27 February 2025 | 0 | – | 330,000 | 210,000 | 120,000 | – |
| 4 March 2025 | 1 | 5 | 200,000 | 80,000 | 120,000 | 16,000 |
| 7 March 2025 | 2 | 3 | 440,000 | 320,000 | 120,000 | 106,667 |
| 13 March 2025 | 3 | 6 | 430,000 | 310,000 | 120,000 | 51,667 |
| 19 March 2025 | 4 | 6 | 280,000 | 160,000 | 120,000 | 26,667 |
| 26 March 2025 | 5 | 7 | 390,000 | 270,000 | 120,000 | 38,571 |
| 30 March 2025 | 6 | 4 | 700,000 | 580,000 | 120,000 | 145,000 |
| 3 April 2025 | 7 | 4 | 460,000 | 340,000 | 120,000 | 85,000 |
| 8 April 2025 | 8 | 5 | 750,000 | 630,000 | 120,000 | 126,000 |
| 12 April 2025 | 9 | 4 | 830,000 | 710,000 | 120,000 | 177,500 |
| 17 April 2025 | 10 | 5 | 710,000 | 590,000 | 120,000 | 118,000 |
| 22 April 2025 | 11 | 5 | 610,000 | 490,000 | 120,000 | 98,000 |
| 27 April 2025 | 12 | 4 | 840,000 | 720,000 | 120,000 | 180,000 |
| 1 May 2025 | 13 | 4 | 780,000 | 660,000 | 120,000 | 165,000 |
| 7 May 2025 | 14 | 6 | 1,010,000 | 890,000 | 120,000 | 148,333 |
| 13 May 2025 | 15 | 6 | 1,070,000 | 950,000 | 120,000 | 158,333 |
| 19 May 2025 | 16 | 6 | 1,040,000 | 920,000 | 120,000 | 153,333 |

(Fig. 4B). Hierarchical clustering of the 500 most differentially abundant genes further separated samples by passage number (Fig. 4C). GO enrichment analysis highlighted functional categories associated with chemotaxis, epithelial cell proliferation, and leukocyte migration (Fig. 4D), driven significantly by the differential abundance of genes such as *Cdc20* and *Cxcl12* (Table 2).

A higher abundance of *Cdc20* transcripts in passage 3 cells may reflect differences in cell-cycle stage or the proportion of dividing cells at the time of RNA collection, given the role of *Cdc20* in mitotic regulation. Increased expression of *Cxcl12* in passage 9 aligns with prior observations that dermal fibroblasts act as a Cxcl12-secreting paracrine cell (Zhang et al., 2024; Janssens et al., 2018; An et al., 2025). The inverse expression profiles of *Cdc20* and *Cxcl12* suggest early shifts toward altered proliferation or signaling profiles with increased passage number. However, we emphasize that bulk RNA sequencing was performed only at passage 3 and passage 9, limiting our ability to define the precise trajectory of these changes. RNA sequencing alone cannot definitively characterize the molecular fate of MEarFs.

MEarFs continued to proliferate through passage 9 without significant enrichment of senescence or quiescence-associated transcripts, especially when compared to immortal T3T-L1 immortal cells and mortal HMLE cell lines (Table 2). In our

hands, the culture remained viable beyond passage 20. While our transcriptomic data represent a significant change in gene abundance for cell taxis, chemokine response, and leukocyte migration, this protocol can be compatible with functional assays, provided that controls and minimal passage numbers are considered. Further studies incorporating single-cell transcriptomics, senescence markers, and functional assays will be required to fully characterize the long-term molecular and cellular adaptations in MEarF cultures.

As cell biology continues to advance, careful attention to model systems remains essential. Reducing sources of variability, such as media composition, passage number, and cell population heterogeneity, is critical for accurate data interpretation and reproducibility. Our MEarF model offers a flexible primary fibroblast applicable to studies of cellular homeostasis, disease mechanisms, host–pathogen interactions, and fibro-adipogenic cell biology. By emphasizing long-term culture stability and minimal media conditions, this protocol provides a cost-optimized and resource-conscious approach to primary fibroblast culture.

## MATERIALS AND METHODS
### Ear tissue collection and cell extraction
**Materials and reagents (vendors and catalog numbers in Table 4)**
- Media (formulation in Table 4), pre-warmed to 37°C
- 100 ml PBS+1% penicillin–streptomycin (P/S)
- Liberase (26–28 Wünsch units/ml)
- Three 10 cm Petri dishes
- Sterile scissors, forceps, and single-edge razor blades
- 15 and 50 ml conical tubes
- 100 µm cell strainers
- 5 ml sterile, single-use syringe
- Pipette aid and 10 ml serological pipettes
- Tissue-culture-treated T25 flasks
- 70% ethanol
- Centrifuge
- $CO_2$- and $O_2$-controlled tissue culture incubator (37°C, 5% $CO_2$)

**Table 4. Names of reagents, vendors, catalog numbers, and final concentrations for minimal media, tissue digestions, and cell freezing**

| Reagent name | Vendor | Concentration (final volume) |
|---|---|---|
| (Minimal) media | | |
| DMEM | Gibco #11960-051 | 87% |
| FBS | Corning #35-010-CV | 10% |
| Antibiotic (P/S) | Gibco #15140-122 | 1% |
| GlutaMAX | Gibco #35050-061 | 1% |
| HEPES | Gibco #15630-080 | 1% |
| MEarF tissue digestion reagent | | |
| Liberase TL Research Grade | Millipore Sigma (Roche) #5401020001 | ~0.13–0.14 Wunsch units/ml |
| MEarF freezing media | | |
| FBS | Corning #35-010-CV | 90% |
| DMSO | ChemCruz #SC-358801 | 10% |

**Notes before starting**
- Prepare at least 100 ml of sterile PBS containing 1% of fresh P/S.
- Prepare 10 ml of PBS+1% P/S stock in a 15 ml conical tube for each pair of ears.
- Ensure that the Liberase aliquot is well reconstituted before use, as it tends to settle during freeze–thaw cycles.
- For the best results, digest one pair of ears (2 cm$^2$) per 15 ml conical tube.

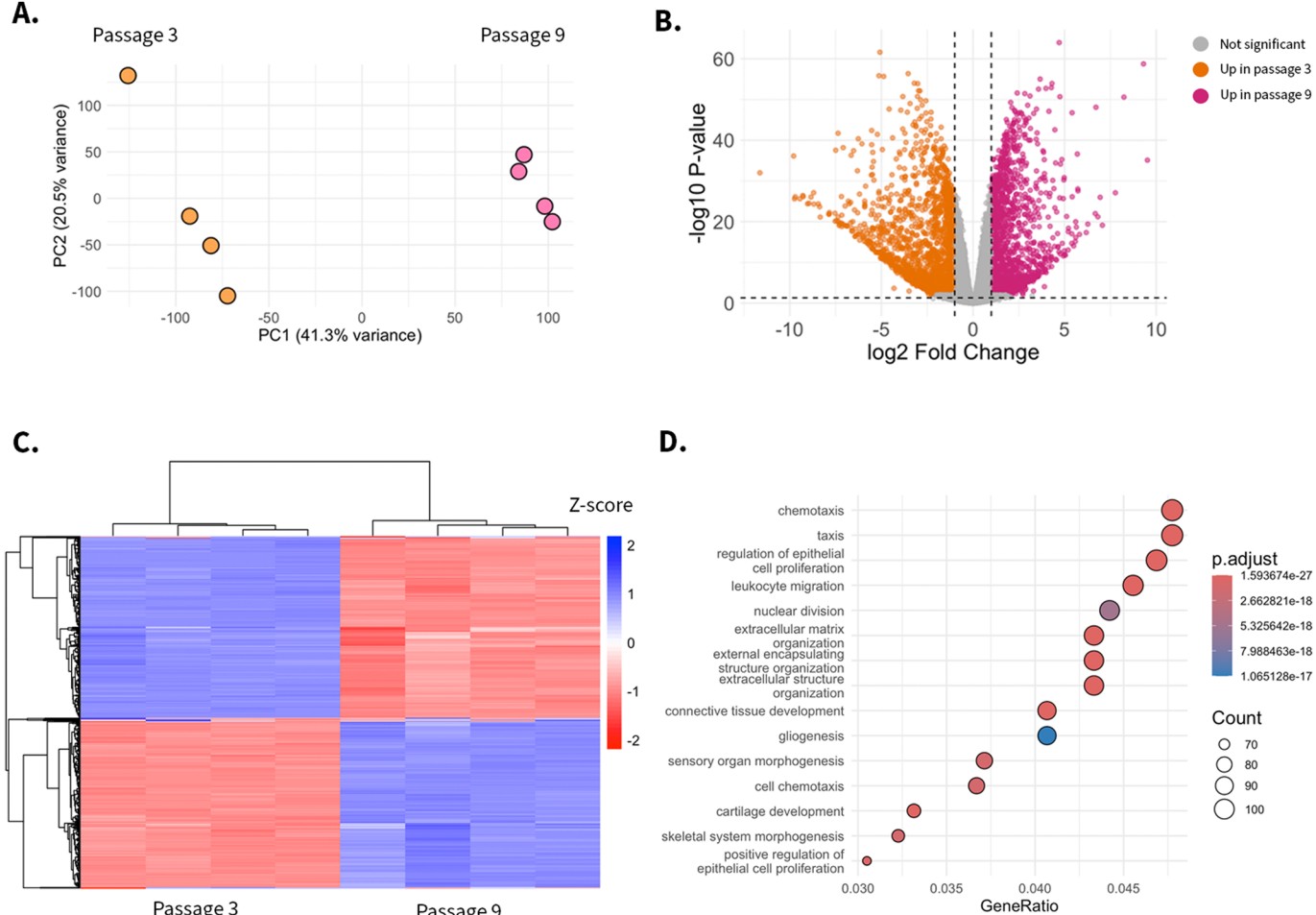

**Fig. 4. Transcriptomic differences between early- and late-passage MEarFs.** (A) Principal component analysis (PCA) of transcript counts from passage 3 (orange, $n$=4) and passage 9 (pink, $n$=4) MEarFs. Samples cluster by passage number with more variability in the early-passage cells. (B) Volcano plot displaying differential gene expression between passage 3 (orange) and passage 9 (pink) MEarFs. Each point represents one mapped gene. Gray points indicate genes with statistically insignificant changes [Benjamini–Hochberg false discovery rate (FDR)>0.05]. Orange points represent genes with higher abundance in passage 3 (log2 fold change>1, FDR<0.05), and pink points represent genes with higher abundance in passage 9 MEarFs (log2 fold change>1, FDR<0.05). Dotted lines indicate significance thresholds. (C) Heatmap of the 500 most differentially abundant genes between passage 3 and passage 9 MEarFs. Hierarchical clustering separates samples by passage number. Color scale represents normalized expression ($Z$-score), with red indicating high expression and blue indicating low expression. (D) Gene ontology (GO) enrichment analysis of differentially expressed genes (adjusted FDR<0.2, |log2 fold change|>1). GO terms are grouped by biological function as determined by the murine genome database (org.Mm.eg.db). Dot size corresponds to the number of genes associated with each term (gene count), and color intensity indicates statistical significance ($P$<0.05).

**Animal preparation and tissue extraction (Fig. 1)**

1. Humanely euthanize the animal following institutional guidelines.
2. In an animal hood, liberally spray the mouse head with 70% ethanol.
3. Using sharp, sterile scissors, remove the mouse ear pinna (outermost ear) by cutting at the junction above the lower ear cartilage. The excised tissue should be approximately 1 cm² per ear (Fig. 1A).
4. Remove excess fur and cartilage with scissors to minimize interference with enzymatic digestion.
5. Using forceps, transfer the tissue to a Petri dish containing 70% ethanol for a brief rinse (Fig. 1B).
6. Transfer the tissue to a second Petri dish containing PBS+1% P/S to rinse off the ethanol. Hold the tissue in this dish while completing extraction from additional animals or for up to 15 min. Clean scissors and forceps with 70% ethanol between each tissue extraction.
7. Once all tissues are collected, transfer them to a third, dry Petri dish, carrying over as little PBS as possible.
8. Using single-edge razor blades in a scissor-like motion, finely mince the tissue, periodically recollecting the tissue into a pile. Continue cutting until the tissue becomes sticky and adheres to the blades.

9. Transfer the minced tissue into the 15 ml conical tube containing 10 ml PBS+1% P/S.
10. If more than one animal is processed, store the 15 ml conical tube on ice until all extractions are complete. Proceed to the next step when all tissue extractions have been made.

**Enzymatic digestion**

11. Add 50 μl of Liberase (~26–28 Wunsch units/ml) to each 15 ml conical tube containing 10 ml PBS+1% P/S and minced tissue, achieving a final concentration of ~0.13–0.14 Wunsch units/ml (Fig. 1C).
12. Rotate samples at 37°C for 90–120 min to begin tissue digestion.
    a. Check samples every 30 min, shaking the tube vigorously to prevent the tissue from adhering to the cap or bottom of the tube. Examine the media for subtle changes in opacity and tissue fragment density over digestion time.
    b. Digestion is sufficiently complete when the PBS is cloudy, and the tissue is opaque with uneven or 'frayed' edges of the tissue chunks. If no tissue chunks remain, digestion has progressed too

far, and the sample should be discarded, as few cells will remain viable (Fig. S1).

## Cell recovery and washing

13. Transfer the 15 ml conical tube containing the digested tissue to a sterile tissue culture hood.
14. Place a 100 µm cell strainer on a new 50 ml conical tube.
15. Using a 10 ml serological pipette, vigorously aspirate to loosen cells still attached to the remaining tissue chunks. Avoid aspirating too high in the pipette to minimize tissue loss on the pipette interior.
16. Transfer the digested tissue suspension through the strainer into the 50 ml conical tube.
17. Rinse the 15 ml conical and serological pipette with at least 10 ml of pre-warmed minimal media, with the rinse falling through the strainer. Equal volumes of minimal media (10 ml) are essential in neutralizing the Liberase enzyme. Additional media can be used throughout the following washing steps 18–22.
18. Use the rubber end of a 5 ml syringe plunger to gently press tissue against the cell strainer to release loosely attached cells from the tissue. Rinse the plunger and strainer with media. Discard the strainer, plunger, and remaining tissue chunks.
19. Centrifuge the 50 ml conical tube with cell suspension at 525 $g$ for 5 min at room temperature.
20. Carefully aspirate the supernatant without disturbing the cell pellet.
21. Resuspend the pellet in 5 ml of warm media.
22. Add 10 ml of media to the conical tube and centrifuge again at 525 $g$ for 5 min to remove any residual Liberase enzyme.
23. Carefully aspirate the supernatant and resuspend the pellet in 5 ml of media.

## Initial culture establishment

24. Transfer the entire resuspended sample into a tissue-culture-treated T25 flask.
25. Incubate at 37°C with 5% $CO_2$.
26. During the first 3 days post-isolation, monitor for major changes in media color. Avoid disturbing the flask, as disturbances early in culture may delay or inhibit cell adherence. If the media turns yellow, assess the flask under light microscopy to determine the next steps.
    a. If bacterial or fungal contamination is visible, sterilize and discard the flask immediately.
    b. If no visible contamination is present, yellowing may indicate overcrowding, and cells should be passaged.
27. Fibroblasts begin adhering within 4–10 days (Fig. 3A).
28. Refresh the media by day 7.
29. By day 14, pass cells at 90–95% confluency, seeding ~120,000 cells per T25 flask.
    a. If cells are not near confluency by day 14, transfer all cells to a new flask.
    b. Timing and seeding density may require optimization for individual laboratory practices. Some mouse genotypes may exhibit different proliferation rates.
30. Pass cells every 4–6 days or whenever the cells reach 90–95% confluency.
31. Track cell counts, passage number, and days between passages to monitor culture health (Table 3).

## Passage, maintenance, and storage of MEarFs

Materials and reagents (vendors and catalog numbers in Table 4)
- Media (formulation in Table 4), pre-warmed to 37°C
- 0.05% Trypsin (Gibco, 25200-056)
- Hemocytometer
- 15 ml conical tubes
- Pipette aid and 2, 5, and 10 ml serological pipettes
- Tissue-culture-treated T25 flask
- 70% ethanol
- Centrifuge

- $CO_2$- and $O_2$-controlled tissue culture incubator (37°C and 5% $CO_2$)
- Logbook or electronic spreadsheet for tracking passage number and cell counts

## MEarF passage protocol

1. Remove media from the T25 flask.
2. Add 2 ml of 0.05% trypsin to detach cells from the flask.
3. Incubate at 37°C for 5 min. Tap the flask at 2 min to assist detachment.
4. Once most cells have detached, add 2 ml of media (equal volume to trypsin for enzyme neutralization).
5. Transfer the cell suspension to a 15 ml conical tube and centrifuge at 250 $g$ for 3 min.
6. Carefully remove the supernatant and resuspend the pellet in 1 ml of media.
7. Count and record the total number of cells using a hemocytometer.
8. Seed ~120,000 cells into a T25 flask with 5 ml of pre-warmed media. Adjust seeding density based on cell behavior and laboratory conditions and practices.
9. Record the passage number, total cell count, and seeding density.
10. Option to freeze cells with the following MEarF freezing protocol.

## MEarF freezing protocol

Materials and reagents (vendors and catalog numbers in Table 4)
- FBS
- DMSO
- 2 ml cryovials (sterile, screwcap)
- Controlled rate freezing container (e.g. Mr. Frosty) or foam tube rack
- −80°C freezer (for initial freezing)
- Liquid nitrogen storage tank (for long-term preservation)

Before starting
- Ensure that the controlled rate freezing container is at room temperature.
- Pre-label cryovials with ethanol-safe markers.

Protocol
1. Complete steps 1–6 of the MEarF passage protocol.
2. After centrifugation, remove the supernatant and resuspend the pellet in 900 µl FBS media.
3. Add 100 µl DMSO to a 2 ml cryovial.
4. Transfer the cell-FBS suspension into the cryovial and mix thoroughly by pipetting.
5. Immediately place in a controlled-rate freezing container (or a foam tube rack) and transfer the cells to −80°C for overnight freezing.
6. After 24 h, transfer cryovials to liquid nitrogen for long-term storage. Cells have been successfully recovered from liquid nitrogen and tested up to 12 months.

## MEarF thawing protocol

1. Media (formulation in Table 4), pre-warmed to 37°C
2. Remove cryovial from storage (liquid nitrogen or −80°C).
3. Rapidly thaw the frozen cell pellet by adding pre-warmed media directly to the cryovial.
4. Transfer all contents of the cryovial to a T25 flask and leave undisturbed in the incubator for at least 3 days. Subsequently, maintain cells as previously described in the passage protocol.

## Ethics statement

Animals were treated in compliance with guidelines set by the Institutional Animal Care and Use Committee (IACUC) of the University of Wisconsin School of Medicine and Public Health (Protocol #M005217), which adheres to regulations and guidelines set by the National Research Council. The University of Wisconsin is accredited by the International Association for Assessment and Accreditation of Laboratory Animal Care.

## Experimental mice

All methods and all experimental protocols were approved by the University of Wisconsin IACUC (Protocol #M005217) as well as the Office of Biological Safety (Protocol #B00000086). C57BL/6 female and male mice, aged 3 to 16 weeks, were humanely euthanized with $CO_2$ and cervical dislocation before ear tissue extraction.

## MEarF culture and media

All cultures were maintained in a 37°C humidified incubator with 5% $CO_2$ and normal oxygen conditions to maintain physiological relevance. MEarFs (isolated in-house, protocol outlined in this paper) were cultured in DMEM (Gibco #11960077) supplemented with 10% FBS (Fisher Scientific #MT35010CV), 1% GlutaMAX Supplement (Gibco #35050061), and 1% P/S (Gibco #15140122). Fresh aliquots of P/S were used for media preparation. Enriched media consisted of Advanced DMEM/F-12 (Gibco #12634010), 15% FBS (listed above), 1% GlutaMAX, and 1% P/S.

## BODIPY stain and Incucyte imaging

MEarFs were cultured to confluence in minimal media or enriched media on a 48-well plate (Thermo Scientific, Nunclon #177224). Live cells were stained with 5 µM BODIPY (Invitrogen #D3922) in minimal media for 30 min at 37°C and then imaged on an Incucyte S3 Live-Cell Analysis Instrument (Sartorius) using phase-contrast and green fluorescence channels with a 20× objective. Images were selected across two independent experiments. Quantification of BODIPY stain was conducted using the Incucyte Sartorius software to detect the number of cells in each media condition, with automatic parameters set to Top-Hat segmentation with a 10 µm radius and 20 GCU threshold and edge split sensitivity of 25. Each media type was grown in separate plates and encompasses eight wells, 16 pictures each. The number of items with green fluorescence per image is tallied as a count. An unpaired $t$-test was used to determine the significance ($P<0.0001$) of the difference in BODIPY between MEarFs grown in enriched media and minimal media. As a negative control, MEarFs imaged without BODIPY stain demonstrated no positive cells outside of background fluorescence (Fig. S2B).

## Immunofluorescent staining

HFFs (ATCC, SCRC-1041), MDCK cells (ATCC, CCL-34), and in-house isolated MEarF cells were seeded on poly-L-lysine German glass coverslips (Electron Microscopy Sciences 72292-02) in four-well plates (Thermo Scientific #144444). On day 5 post-seeding, cells were fixed with 500 µl of 4% formaldehyde (Thermo Scientific #28908) for 30 min at room temperature, covered from light. After fixation, the cells were gently rinsed with 500 µl PBS three times to rinse off residual formaldehyde. Cells were permeabilized with 500 µl of PBS+0.1% Triton in each well for 5 min at room temperature on a rocker, covered from light, then rinsed three times with PBS. Nonspecific proteins were blocked with 500 µl of PBS containing 5% BSA (Sigma-Aldrich #A9418-10G) for 30 min at room temperature, on a rocker, covered from light. Primary vimentin antibody (R&D Systems #MAB2105-SP) was added immediately after removal of the BSA solution at a concentration of 1:250 µl PBS and incubated overnight at 4°C. The following day, cells were rinsed five times with PBS and incubated with goat anti-rat IgG (H+L) cross-adsorbed secondary antibody conjugated to Alexa Fluor 584 (Invitrogen #A-11006) at a concentration of 1:250 µl PBS for 2 h, in the dark, at room temperature. Nuclei were stained with DAPI (Sigma-Aldrich #D9564-10MG) at a concentration of 1:1000 µl PBS for 20 min at room temperature. Coverslips were carefully removed from the wells, rinsed in a beaker of DI water, and mounted cell-side down onto microscope slides with 4 µl of VECTASHIELD Vibrance Antifade Mounting Medium (Vector Laboratories #H-1700-2). Coverslips were sealed onto the microscope slides with nail polish, stored at 4°C, and imaged on a Zeiss Axioplan III motorized microscope with a 40× air objective. Photos were adjusted in Adobe Photoshop by linearly and uniformly adjusting the curve for image clarity. To confirm antibody specificity, secondary-only controls were performed for each cell type following the staining protocol stated above but omitting the primary vimentin antibody (Fig. S3). Brightness and contrast for all immunofluorescence images were adjusted in Adobe Photoshop by linearly and uniformly

repositioning the curve points to improve visualization of the fluorescent signal.

## Cell preparation for RNA isolation

After the second passage post-isolation, an aliquot of MEarFs was cryopreserved following the MEarF freezing protocol outlined in this article. A separate aliquot from the same isolation was continuously cultured. When the ongoing culture of MEarFs reached passage 6, we thawed the frozen stock of the same cell line and cultured both populations in parallel for two more passages. At passages 3 and 9, we seeded 40,000 cells into four replicate wells of a six-well plate (Thermo Scientific Nunc #140675) with 3 ml of media. The following day, an additional 2 ml of media was added to each well. On day 4 post-seeding, cells were lysed and detached from the plate with a 15-min treatment of 1 ml TRIzol (Invitrogen #15596018). Cells were then stored at −80°C overnight. The next day, RNA of the MEarF cell culture monolayer was extracted using phenol-chloroform separation and isopropanol precipitation. RNA samples were treated with DNase 1 (New England Biolabs #M0303S) and tested for RNA yield, and nucleic acid purity was confirmed by the A260/A280 and A260/A230 ratios. The University of Wisconsin-Madison Biotechnology Center's Gene Expression Center Core Facility [research resource identifier (RRID): SCR_017757] conducted quality control assessments. RNA library preparation, sequencing, and read demultiplexing were performed by the University of Wisconsin Biotechnology Center DNA Sequencing Facility (RRID: SCR_017759). The Illumina TruSeq-Stranded mRNA kit was used for poly(A)$^+$ enrichment for library preparation. Double-stranded cDNA was purified using NEB SPRIselect beads. Library quantification was performed using the PicoGreen assay on an Agilent BioTek Synergy H1 microplate reader, and library quality was assayed on the 4200 Tape Station on a D1000 ScreenTape. After passing quality control, the samples were sequenced on the Illumina NovaSeq X Plus 10B platform with 2×150 base-paired-end reads. Each sample generated approximately 50 million paired-end reads with an average read length of 150 bp per sample (http://www.bioinformatics.babraham.ac.uk/projects/fastqc/).

## RNA-sequencing analysis

RNA-sequencing reads were processed to remove low-quality reads using Trimmomatic (v0.39) (Bolger et al., 2014). Reads were then aligned to the *Mus musculus* reference genome (GRCm39, https://useast.ensembl.org/) using the Spliced Transcripts Alignment to a Reference Program (STAR, v2.7.5c) (Dobin et al., 2013). Default STAR parameters were used except for the following: maximum mismatch of 2 bp, and intron length range set to 20–100,000 bp (Dobin and Gingeras, 2015). Quantification of mapped reads and generation of a counts table were performed (RSEM, v1.3.1) (Li and Dewey, 2011). Counts were imported into R (tximport, v1.36.1), and differential expression analysis was conducted (DESeq2, v1.48.2) (Love et al., 2014; Soneson et al., 2015). Normalized counts of MEarF monolayers at passage 3 and passage 9 were compared to representative and previously published datasets from RStudio resources. Representative epithelial and mesenchymal transcriptomes were collected from HMLE cells before ($n=3$) and after ($n=3$) epithelial–mesenchymal transition. HMLE data were procured from the ExperimentHub package and the emtdata library in RStudio. The dataset represent 27,515 genes across ten samples. Similarly, T3T-L1 data were collected from the RStudio package, curatedAdipoRNA. Pre-adipocyte cells were qualified as timepoint hour 0 ($n=22$), and adipocyte-like cells were qualified by the last possible timepoint, 168 h ($n=25$). DESeq2 analysis of differentially expressed genes of all data points was done in parallel. Select genes and gene markers are represented and outlined in this article (Fig. 3C, Table 2). Log-transformed values were used to generate PCA plots. Volcano plot (ggplot2, v4.0.0) was created using a matrix of differentially expressed genes, with significance determined by a Benjamini–Hochberg false detection rate (FDR) of 0.05. Hierarchical clustering (pheatmap v1.0.13) of the top 500 most differentially abundant genes and sample clustering using Euclidean distance. GO enrichment analysis was performed (clusterProfiler v4.16.0, enrichGO). GO terms were represented by all three functional gene categories, with most genes mapping to the biological process (2642 genes) and fewer to cellular

component (163 genes) and molecular function (276 genes). We applied a Benjamini–Hochberg *P*-value cutoff of 0.05 and an adjusted FDR threshold of 0.02. GO enrichment was conducted on genes with assigned Entrez Gene Identifiers using the org.Mm.eg.db (https://bioconductor.org/packages/release/data/annotation/html/org.Mm.eg.db.html) annotation package. Broad functional categories and lenient cutoff values were used for exploratory analysis.

## Statistical analysis

Data were analyzed using R software integrated within each analysis package, with cutoff values as specified and listed above.

## Acknowledgements

The authors thank the shared campus resources with the University of Wisconsin-Madison Biotechnology Center Gene Expression Center (RRID: SCR_017757) for RNA library preparation and building. We also thank the University of Wisconsin-Madison Biotechnology Center DNA Sequencing Core Facility (RRID: SCR_017759) for running and demultiplexing RNA sequences. We thank the current and former members of the Knoll Lab for their support and editorial contributions.

## Competing interests

The authors declare no competing or financial interests.

## Author contributions

Funding acquisition: L.J.K.; Investigation: K.L.B., B.J.E.; Methodology: K.L.B., N.M.D., B.J.E., K.M.C., E.H.B.; Project administration: L.J.K.; Supervision: N.M.D., L.J.K.; Visualization: K.L.B., E.H.B.; Writing – original draft: K.L.B., B.J.E.; Writing – review & editing: K.L.B., N.M.D., B.J.E., K.M.C., E.H.B., L.J.K.

## Funding

Funding for this project was provided by the Food Research Institute (L.J.K.) and the National Institutes of Health (1R01AI172874 to L.J.K., T32AI55397 to K.L.B., T32AI007414 to B.J.E., and F32362AI172084 to N.M.D.). Open Access funding provided by National Institutes of Health (1R01AI172874). Deposited in PMC for immediate release.

## Data and resource availability

RNA-sequencing data are available under BioProject ID PRJNA1400943. Gene counts and DESeq2 output files are posted with the raw RNA-sequencing data and can be accessed freely through NCBI Gene Expression Omnibus (GSE330836). All other relevant data and details of resources can be found within the article and its supplementary information.

## First Person

This article has an associated First Person interview with the first author of the paper.

## Peer review history

The peer review history is available online at https://journals.biologists.com/bio/lookup/doi/10.1242/bio.062483.reviewer-comments.pdf.

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
