## [Peer Review File · Biology Open]

Isolation, Validation, and Long-Term Culture of Mouse Ear Fibroblasts

Katie L. Barnes, Nicole M. Davis, Billy J. Erazo, Katie M. Cataldo, Emmaly H. Bertges and Laura Knoll
10.1242/bio.062483

Editor: Alissa Armstrong

Review timeline

Original submission:	11 January 2026
Editorial decision:	15 January 2026
First revision received:	10 April 2026
Accepted:	17 April 2026

Original submission

First decision letter

MS ID#: bio.062483

MS Title: Isolation, Validation, and Long-Term Culture of Mouse Ear Fibroblasts

Authors: Katie L. Barnes, Nicole M. Davis, Billy J. Erazo, Katie M. Cataldo, Emmaly H. Bertges and Laura Knoll

I have now reached a decision on the above manuscript.

The reviewer reports are shown at the bottom of this email.

As you will see, the reviewers are enthusiastic about this technique and think it will be a valuable resource; however, they have raised a number of substantial criticisms that prevent me from accepting the paper at this stage.

They suggest, however, that a revised version might prove acceptable, if you can address their concerns. If you think that you can deal satisfactorily with the criticisms on revision, I would be pleased to see a revised manuscript. We would then return it to the reviewers.

At this stage, we also ask you to ensure your manuscript complies with our formatting guidelines. Provided you are able to fully address the referees' comments, we are positive about publication of your paper (we accept over 95% of revision submissions) and therefore hope you won't mind any extra work involved in reformatting your manuscript at this point.

Please upload both a 'clean' version of your Word file, along with a highlighted version clearly showing where you have made changes in the revised manuscript. Please avoid using 'Track changes' in Word files as these are lost in PDF conversion.

I should be grateful if you would also provide a point-by-point response detailing how you have dealt with the points raised by the reviewers in the 'Response to Reviewers' box. Please attend to all of the reviewers' comments. If you do not agree with any of their criticisms or suggestions please explain clearly why this is so.

Reviewer 1

Comments for the author

This manuscript presents a streamlined, cost-effective protocol for isolating and maintaining mouse ear fibroblasts (MEarFs) long-term. The authors address a genuine gap in the field, the lack of standardized, reproducible primary fibroblast culture methods that minimize contamination and support extended passaging. The work is timely, well-motivated, and generally well-executed. The inclusion of transcriptomic profiling adds depth and strengthens the manuscript's impact. However, several issues require clarification or additional controls. Some methodological details are incomplete, certain claims are not fully supported by the data presented, and a few irregularities in the figures and text should be addressed before publication. Addressing these issues would substantially strengthen the rigor, clarity, and reproducibility of the work.

Major Comments

1. BODIPY staining lacks unstained controls, and vimentin IF lacks secondary-only controls. The MDCK negative control appears too dim to convincingly demonstrate absence of vimentin.
2. Figures 2 and 3 do not specify the number of biological replicates, making it difficult to assess reproducibility.
3. TPMs are not suitable for t-tests. Authors should use normalized counts (e.g., DESeq2 outputs) or log-transformed counts with appropriate statistical justification.
4. The conclusion that minimal media "prevents" contamination is based solely on qualitative imaging. Quantification (e.g., % BODIPY+ cells) is needed.
5. Transcriptomic analysis was performed only at passages 3 and 9, yet the manuscript claims fibroblast identity is maintained through passage 13. This limitation should be explicitly acknowledged.
6. The rationale for culturing at 3% O₂ is not provided; digestion-endpoint criteria are subjective; and the choice of Liberase over collagenase/dispase is not justified.
7. The discussion of fibro/adipogenic progenitors and media-dependent fibroblast phenotypes is limited and should be expanded.

Minor Comments

1. Clarify whether ear tissue from younger mice yields different long-term outcomes, since this is mentioned but not explored.
2. Improve clarity of MDCK negative control images; current presentation is difficult to interpret.
3. Provide objective indicators of "proper digestion" beyond visual descriptors.
4. Ensure all figure panels include scale bars and consistent labeling.
5. Expand the limitations section to explicitly state untested assumptions (e.g., strain-specific differences, long-term senescence markers not assessed).
6. Consider adding a brief note on whether this protocol is compatible with downstream functional assays (migration, contraction, cytokine response).

Reviewer 2

Comments for the author

Overall, the study by Barnes et al., provides a useful, effective protocol for isolating fibroblasts from mouse ear samples. They provide excellent detail on how to undertake this protocol, and some useful verifications that highlight the purity of the isolation and importance of fibroblast passage number. I have a few minor concerns regarding presentation and analysis of data, and some additional suggested controls/analyses, which, if performed, will provide this study with wider relevance and extra rigour. Below are my comments following the BiO reviewing rubric.

1. Experimental quality

a. Does each figure have the proper controls?

In general, yes, however additional controls would increase the robustness of the manuscript and further support the conclusions drawn. Addition of an epithelial control (such as MDCK cells as used in Figure 2), and a fibroblast cell line that can be a bit "adipocyte like" such as 3T3-L1 cells" for validating key transcript markers would be useful. Currently while there are TPM numbers presented for selected transcripts, without an epithelial control for reference these are hard to

interpret. For example, is ~95 TPM a lot of CDC20a transcript, or negligible? A point of reference is required.

I understand transcriptomic analysis of additional cell lines would be excessive to achieve this. Targeted alternate experiments such as qPCR or Western blot (or another method) validation of the markers presented in Figure 3 (vimentin, CDC20a, FoxC2, normalised to GAPDH), and adipocyte markers selected from Table 2, could be performed in the isolated fibroblasts at P3 and P9 in minimal media, isolated fibroblasts from enriched media, epithelial and fibroblast cell lines. This would provide extra validation that the presented protocol does indeed deliver specific isolation of fibroblasts that can be cultured for multiple passages without developing strong characteristics of epithelial and adipocyte-like cell lines.

Alternative to undertaking additional experiments, if transcriptomic/expression data exists in the literature that would provide an indication of the expression of these markers in epithelial/adipocyte cell lines or equivalent, this data should be discussed or referenced and ideally presented in a table providing comparison to the current data.

b. Are experiments performed using appropriate methods that will answer the question (or test the hypothesis or support the observations) posed by the authors? Is the right tool used for the job? All method used is appropriate for testing the protocol established in this manuscript.

c. Were the data analysed using appropriate statistical tests?

Data were analysed using appropriate statistical tests. However, as detailed below in 2b, images in Figure 2 and Figure 3 were not analysed and therefore not subjected to statistical testing. This should be rectified as suggested below.

2. Reproducibility

a. Were experiments in each figure performed using adequate number of biological replicates?

An adequate number of replicates were used for presented data. However, there were some instances where results were referred to but data not presented. Instances of this were in: Discussion lines 182/183 in reference to isolating cells from smaller tissue inputs; Discussion lines 194-196.

In Discussion lines 182/183, no data is presented (unless I missed it) of fibroblasts isolated from ear hole punches despite the assertion that this was performed successfully. Given the ethical and experimental advantages of this approach, at least presenting a few images to complement those in Fig 3A or Fig 2 C-E would be good to show that such an isolation produces cells with similar morphology to the larger ear isolation.

In Discussion lines 194-196, there is no data to complement the statement about isolation of fibroblasts from a 3-week-old mouse and that adipocyte-like cells were still derived from this isolation. This would be a useful supplementary figure.

b. Is there sufficient raw data to assess the rigor of the analysis?

For transcript related figures (3, 4, Table 2) sufficient raw data is presented, and RNA sequencing data will most likely be available on NCBI (there is a BioProject ID, but the ID does not yet seem to be active).

For imaging related figures (Figure 2, Figure 3A, B) insufficient raw data is presented, which is especially important given there is no quantification of these images. These images should be quantified where possible. Quantification of the percentage of BODIPY positive cells in Fig 2A,B should be performed. A blinded, manual count of adipocyte like cell formation in Fig 2C-E should be undertaken. The mean vimentin intensity per cell should be quantified in Fig 3B. Where there is an absence of image quantification, a panel of images of each condition from multiple biological repeats needs to be presented in a supplementary figure.

c. Does the methods section provide sufficient detail to permit reproducibility?

The methods are excellent, especially the step-by-step protocol of how to isolate the fibroblasts. The tables and supplementary figures to complement the methods are very useful as well.

3. Completeness

a. Are the author's conclusions supported by the data?

For the most part, yes. However, as stated in 1a, additional controls or references to transcript data in literature would strengthen the study and the ability of the authors to assert that the isolation protocol produces a relatively pure fibroblast culture.

A minor point, but the statement related to Figure 4A in the figure legend "Samples cluster by passage number with minimal variation" is a bit strong, given that one of the P3 isolates is very different looking from the other 3 P3 isolates in the PC analysis. It could be useful to highlight that for the most part biological replicates cluster together, but the apparent P3 outlier highlights the need to perform multiple replicates as outliers may have very different cell characteristics.

b. Are there any flaws in the experimental design that invalidate the approach taken by the authors?

There are no experimental design flaws that invalidate the study or approaches taken.

c. Are there experiments that have not been performed, but if true would disprove the conclusion? If yes, and if such experiments would be costly or time-consuming to perform, do the authors acknowledge this in a discussion of the limitations?

While I do not think results from the experiments/analysis of transcripts from other cell lines in literature I suggested in 1a would invalidate the study, they would strengthen the authors ability to conclude their isolation method produces a pure primary fibroblast line. I do not think performing the suggested experiments would take an excessive time to perform, however as I suggested, a non-experimental alternative would be to compare to published transcriptomic studies on epithelial and adipocyte-like cell lines.

4. Scholarship

a. Do the authors cite and discuss the merits of relevant data that would argue against their conclusion?

The authors provide balanced citations of publications that do not necessarily support their findings and provide potential explanations for why their results may differ from those published. A table (Table 1) is also presented summarising findings of other published fibroblast isolation protocols. A comment on whether or not there is any myocyte contamination in the current protocol, as was observed for Bravo et al (presented in Table 1) would be useful (and perhaps speculation as to why the authors current protocol does not seem to have myocyte contamination).

b. Do the authors cite and discuss the merits of relevant data that would support their conclusion? Supporting literature is discussed, both in text and presented in Table 1.

c. For techniques/methods manuscripts, Do the authors cite and discuss the current state of the field and clearly explain how the method improves the field?

The authors discuss the current state of the field well for the most part. While the authors make statements about how their method improves the field, some of these statements need additional discussion or supporting evidence.

That this protocol is "cost effective" is stated numerous times, while in other instances the authors say "cost-optimized". "Cost-optimized" should be consistently used because as far as I can tell, the only cost reduction over many other protocols is through using less serum, using 10% FBS compared to 15% FBS in some- but not all- other protocols. For example, Ningshih et al also used 10% FBS and standard DMEM but also used only mechanical disruption of tissue- this would surely be more cost effective than the authors current protocol which includes an enzymatic digestion step, with the added cost of the enzyme. I assume enzymatic digestion helps with the yield and purity of the isolation, hence it being a "cost-optimized" protocol. The authors could provide comment on why the liberase digestion aids the protocol.

The authors also state their protocol prevents non-fibroblast contamination, which it clearly does regarding adipocytes. However, given other protocols they have cited found myocyte contamination of their fibroblast cultures, they authors should comment on the potential for such contamination in their protocol, and why their protocol avoids this contamination. Providing more phase contrast images as in Figure 2A and B in a supplementary figure to show there is minimal myocyte contamination across a large number of cells/ fields of view would provide a useful addition to the paper as well.

Reviewer's Responses to Questions

Experimental quality

Does each figure have the proper controls?

If 'No', please indicate reasons in Comments for Author box below.

Reviewer #1:

- No

Reviewer #2:

- Yes

Were the data analyzed using appropriate statistical tests?

If 'No', please indicate reasons in Comments for Author box below.

Reviewer #1:

- No

Reviewer #2:

- Yes

Reproducibility

Were experiments performed using adequate number of biological replicates?

If 'No', please indicate reasons in Comments for Author box below.

Reviewer #1:

- No

Reviewer #2:

- Yes

Does the methods section provide sufficient detail to permit reproducibility?

If 'No', please indicate reasons in Comments for Author box below.

Reviewer #1:

- Yes

Reviewer #2:

- Yes

Completeness

Are the manuscript's conclusions supported by the data?

If 'No', please indicate reasons in Comments for Author box below.

Reviewer #1:

- Yes

Reviewer #2:

- Yes

Scholarship

Do the authors cite and discuss the merits of data that would argue for and against their conclusion?

If 'No', please indicate reasons in Comments for Author box below.

Reviewer #1:

- Yes

Reviewer #2:

- Yes

Does the manuscript title & abstract accurately reflect the contents of the manuscript, without hyperbole?

If 'No', please indicate reasons in Comments for Author box below.

Reviewer #1:

- Yes

Reviewer #2:

- Yes

First revision

Author response to reviewers' comments

Dear Dr. Armstrong, Handling Editor *Biology Open*,

Thank you for the opportunity to improve our manuscript. We appreciate the time that our reviewers took to thoroughly analyze our manuscript and offer great suggestions. We have addressed their critiques on a point by point basis (our response is in yellow). In this resubmission we have included a copy of the manuscript with the changes highlighted. Thank you again, and we hope this manuscript is now suitable for publication in *Biology Open*.

Reviewer 1: Major Comments

1. BODIPY staining lacks unstained controls, and vimentin IF lacks secondary only controls. The MDCK negative control appears too dim to convincingly demonstrate absence of vimentin. Excellent points. Unstained controls for BODIPY staining are included in Figure S2B, and Vimentin IFA secondary controls for MEaFs, HFFs, and MDCKs are presented in Figure S3.

Brightness and contrast for all immunofluorescence and light microscope images were linearly and uniformly adjusted in Adobe Photoshop to enhance visualization of the fluorescence signal in Fig. 3 and Fig. S3. Details for the adjustments are outlined in the methods section for immunofluorescent staining: "Brightness and contrast for all immunofluorescence images were adjusted in Adobe Photoshop by linearly and uniformly repositioning the Curve points to improve visualization of the fluorescent signal."

2. Figures 2 and 3 do not specify the number of biological replicates, making it difficult to assess reproducibility.

Thank you for your comment. The number of biological replicates is important to include. In Figure 2A-B, we now show additional representative images of BODIPY-stained MEaFs, across 3 independent experiments, clarified in the Methods- BODIPY stain and Incucyte Imaging. The photos in Figure 2C-E, now labeled D-F, are representative of 2 separate experimental flasks of MEaF cultures. The number of replicates for Fig. 2 and 3 are now noted in the methods text and figure legends and additional representative images are included for all Incucyte and IFA images in Fig. 2, 3, S2, and S3. Figure 3A now also includes additional representative images of Vimentin staining of the MEaFs, HFFs, and MDCKs. Additional photos for the morphological progression of MEaFs in Figure 3B can be found in Figure S4.

For transcriptomic comparisons, Passage 3 (P3) and Passage 9 (P9) MEaFs n = 4. Pre (n = 22) and post (n = 25) adipocyte differentiation of T3T-L1s, and Human Mammary Epithelial (HMLE) cells pre (n = 3) and post (n = 3) epithelial transition to mesenchymal cells, with more details outlined in the Methods section for RNA sequencing analysis and under the following comment.

3. TPMs are not suitable for ttests. Authors should use normalized counts (e.g., DESeq2 outputs) or logtransformed counts with appropriate statistical justification.

We appreciate this reviewer's insight into appropriate comparisons and statistics for transcriptomics. We updated figure 3C with a new formatted graph to appropriately and clearly represent transcript abundance of passage 3 and passage 9 MEaFs compared to published data sets of adipocyte-like T3T-1L cells, and HMLE mesenchymal, and epithelial data. Normalized counts

were calculated with DESeq2 and analyzed in parallel for downstream transcriptomic comparisons. In collaboration with Reviewer #2's request for transcript comparisons with other cell types, figure 3C has been updated. The methods are excerpted here:

“Normalized counts of MEarF monolayers at passage 3 and passage 9 were compared to representative and previously published datasets from RStudio resources. Representative epithelial and mesenchymal transcriptomes were collected from Human Mammary Epithelial (HMLE) cells before (n = 3) and after (n = 3) epithelial-mesenchymal transition. HMLE data were procured from the Experimenthub package and the emtdata library in RStudio. The dataset represents 27,515 genes across 10 samples. Similarly, T3T-L1 data were collected from the RStudio package, curatedadipodata. Preadipocyte cells were qualified as timepoint hour 0 (n = 22), and adipocyte-like cells were qualified by the last possible timepoint, 168 hours (n = 25). DESeq2 analysis of differentially expressed genes of all data points were done in parallel. Select genes and gene markers are represented and outlined in this manuscript (Fig. 3C, Table 2).”

4. The conclusion that minimal media "prevents" contamination is based solely on qualitative imaging. Quantification (e.g., % BODIPY+ cells) is needed.

We agree that quantitative analysis would be necessary to claim prevention of contamination, so we have updated the language to now read, “Minimal media reduces adipocyte-like contamination in MEarF cultures.” We included three Incucyte photos of MEarFs grown in enriched or minimal media to show the stark difference of cells stained with BODIPY lipid dye between media conditions in Figure 2A and B. Quantification of green fluorescence in BODIPY stained MEarFs grown minimal media and enriched media are now provided in Figure 2C. The details on the quantification are outlined in the methods section:

“Quantification of BODIPY stain was conducted using the Incucyte Sartorius software to detect the number of cells in each media condition with automatic parameters set to Top-Hat segmentation with a 10um radius and 20 GCU threshold and edge split sensitivity of 25. Each media type was grown in separate plates, and encompasses 8 wells, 16 pictures each. The number of items with green fluorescence per image are tallied as a Count. An unpaired t-test was used to determine the significance ($p < 0.0001$) of the difference in BODIPY between MEarFs grown in enriched media and minimal media.”

5. Transcriptomic analysis was performed only at passages 3 and 9, yet the manuscript claims fibroblast identity is maintained through passage 13. This limitation should be explicitly acknowledged.

We appreciate the reviewer pointing out the inconsistency with our wording. The transcriptomics data support fibroblast identity to passage 9. In Figure 3, we used morphologic profiling to confirm fibroblast identity to passage 13, which does conflate two experimental conclusions. To clarify our findings, we have updated this statement to explicitly acknowledge the limitations of our claim.

First in the Introduction:

“In contrast, minimal media (DMEM + 10% FBS) proved sufficient for both isolation and long-term culture, with fibroblast gene expression confirmed through passage 9.”

And in the Methods- Minimal media reduces adipocyte-like contamination in MEarF culture:

“Cells maintained exclusively in minimal media did not show a noticeable population of adipocyte-like cells through passage 13, as determined by morphology under light microscopy, indicating that growth in minimal media can reduce adipocyte-like cell contamination in MEarFs culture.”

6. The rationale for culturing at 3% O₂ is not provided; digestion endpoint criteria are subjective; and the choice of Liberase over collagenase/dispase is not justified.

Thank you for catching the 3% O₂ error, it is indeed incorrect, we used normoxic conditions for growth conditions. We are working on a hypoxia manuscript and added the incorrect conditions. Thank you again for catching this error.

We agree that the digestion endpoint is subjective. The text in the methods section for Enzymatic Digestion now reads:

“Digestion is sufficiently complete when the PBS is cloudy and the tissue is opaque with uneven or “frayed” edges of the tissue chunks. If no tissue chunks remain, digestion has progressed too far, and the sample should be discarded, as few cells will remain viable (Fig. S1).”

The figure legend of Figure S1 also includes further elaboration:

“Circles highlight representative tissue pieces demonstrating changes in tissue architecture and opacity during digestion. At 30 minutes, tissue edges and color remain intact. By 60 minutes, tissue edges appear frayed. At 90 minutes, optimal digestion is achieved when tissue edges are extensively frayed with visible color change, and the solution appears cloudy. Over-digestion results in the complete loss of visible tissue fragments.”

We also updated supplemental figure 1 to more clearly highlight the subtle changes for digestion with circles and text description emphasizing the conditions of the tissue to define the endpoint.

Liberase was our digestion enzyme of choice because Liberase has been reported to most effectively extract cells in comparison to other enzymes (Brandhorst et al., 2010). This rationale is now explicitly described in the Results section- Streamlined isolation of primary fibroblasts from mouse ear tissue:

“Liberase was chosen for enzymatic tissue disruption because of the documented success of tissue degradation as a mix of Collagenases I, II, and proteases (Brandhorst et al. 2010).”

7. The discussion of fibro/adipogenic progenitors and media dependent fibroblast phenotypes is limited and should be expanded.

We thank the reviewer for their interest in the fibro-adipogenic progenitors. We are also excited about these findings and agree that we should expand on them in the discussion to bring attention to the understudied population of primary adipo-progenitor cells. We drew interesting connections between the enriched MEArF media environment and that fibro-adipogenic cells thrive.

“FAP cells are heterogeneous and serve as key regulators of muscle regeneration and stem cell differentiation (Molina et al., 2021). Importantly, the presence of FAP cells is an indicator of culture stress and dysregulation as markers of wound healing (Smith and Loomis, 2023). Other primary fibroblast isolation protocols (Table 1) characterize cells only at early passages, whereas we observed the cells across multiple passages. We hypothesize that FAP cells proliferated in our cultures due to enriched media conditions and prolonged culture stress. This suggests our protocol could be adapted to intentionally establish FAP cell models, providing a platform for studying in vitro fibro-adipogenic differentiation.”

Minor Comments

1. Clarify whether ear tissue from younger mice yields different longterm outcomes, since this is mentioned but not explored.

Thank you for this suggestion. We did not perform long term outcomes and transcriptomics on cells isolated from younger mice because the hypothesis tested in that experiment was that MEArFs from younger mice would have little to no adipocyte-like cells. This hypothesis was incorrect, but we did notice from two independent isolations, cells isolated from younger mice reached confluency more rapidly compared to cells isolated from older mice for the first passage (Figure S4). After passage 1, the MEArFs isolated from both younger mice (3 weeks old) and older (12 weeks old) mice grew equally and validated through morphological characterization with light microscopy (Figure S4). We clarified the discussion to say:

“Donor age was hypothesized to contribute to the appearance of differentiating adipocyte cells, so we isolated MEArFs from 3-week-old mice relative to 12-week-old adult mice (Berry et al., 2013; Bruder et al., 2022). When grown in enriched media, our unpublished observations led to MEArFs from young donors still producing adipocyte-like cells, suggesting that donor age alone is insufficient to prevent co-isolated cell contaminants. In contrast, MEArFs isolated in minimal media conditions, from donors of 3-week, 6-week, and 12-week-old mice, reached confluence in minimal media without adipocyte contamination (Fig. S4). This suggests that media composition, not the age of the donor, is likely responsible for the proliferation of adipocyte-like contaminants. We observed that the only donor age-related differences to be young donors reached confluence post-isolation sooner than the older donors, but beyond passage 0, donor age did not significantly influence confluency rate.”

2. Improve clarity of MDCK negative control images; current presentation is difficult to interpret. We agree, so as discussed above, we increased the color and contrast on a uniform and linear scale in Photoshop for all IFA and light microscope images to increase their clarity.

3. Provide objective indicators of "proper digestion" beyond visual descriptors. We agree so "proper digestion" is now described as "sufficient digestion", visual indicators were elaborated, and a representative piece of tissue were added to figure S1. The text in the results section and figure legend reads as stated above in Comment #6.

4. Ensure all figure panels include scale bars and consistent labeling. We apologize that there was inconsistency in figure panels regarding scale bars and labeling. These have been corrected throughout all of the figures.

5. Expand the limitations section to explicitly state untested assumptions (e.g., strain-specific differences, long-term senescence markers not assessed). Excellent point for us to clarify. The discussion section We have now added a paragraph in the discussion to describe the limitations of our study. "MEarFs continued to proliferate through passage 9 without significant enrichment of senescence or quiescence-associated transcripts (Table 2), and in our hands, remained viable beyond passage 20. While our transcriptomic data represent a significant change in gene abundance for cell taxis, chemokine response, and leukocyte migration, this protocol can be compatible with functional assays, provided that controls and minimal passage numbers are considered. Further studies incorporating single-cell transcriptomics, senescence markers, and functional assays will be required to fully characterize the long-term molecular and cellular adaptations in MEArF cultures."

We did note that adipocyte-like cells appeared in five different C57BL/6 genotypes and was observed by two independent researchers in our lab. This is clarified in the results- Minimal media reduces adipocyte-like contamination in MEArF culture:

"Cells with an accumulation of lipid-like structures are indicative of adipocytes and were found in MEArF cultures in enriched media, regardless of the donor's genotype (Fig S2A)."

In the discussion, we included analysis of quiescence and senescence markers of MEArFs compared to the benchmark samples analyzed, as represented in Table 2. "MEarFs continued to proliferate through passage 9 without significant enrichment of senescence or quiescence-associated transcripts, especially when compared to immortal T3T-L1 immortal cells and mortal HMLE cell lines (Table 2)."

6. Consider adding a brief note on whether this protocol is compatible with downstream functional assays (migration, contraction, cytokine response).

Excellent suggestion, we have added the following statement in the discussion: "While our transcriptomic data represent a significant change in gene abundance for cell taxis, chemokine response, and leukocyte migration, this protocol can be compatible with functional assays, provided that controls and minimal passage numbers are considered."

Reviewer 2: Overall, the study by Barnes et al., provides a useful, effective protocol for isolating fibroblasts from mouse ear samples. They provide excellent detail on how to undertake this protocol, and some useful verifications that highlight the purity of the isolation and importance of fibroblast passage number. I have a few minor concerns regarding presentation and analysis of data, and some additional suggested controls/analyses, which, if performed, will provide this study with wider relevance and extra rigour. Below are my comments following the BiO reviewing rubric.

1. Experimental quality

a. Does each figure have the proper controls?

In general, yes, however additional controls would increase the robustness of the manuscript and further support the conclusions drawn. Addition of an epithelial control (such as MDCK cells as used in Figure 2), and a fibroblast cell line that can be a bit "adipocyte like" such as 3T3-L1 cells for validating key transcript markers would be useful. Currently while there are TPM numbers presented for selected transcripts, without an epithelial control for reference these are hard to

interpret. For example, is ~95 TPM a lot of CDC20a transcript, or negligible? A point of reference is required.

I understand transcriptomic analysis of additional cell lines would be excessive to achieve this. Targeted alternate experiments such as qPCR or Western blot (or another method) validation of the markers presented in Figure 3 (vimentin, CDC20a, FoxC2, normalised to GAPDH), and adipocyte markers selected from Table 2, could be performed in the isolated fibroblasts at P3 and P9 in minimal media, isolated fibroblasts from enriched media, epithelial and fibroblast cell lines. This would provide extra validation that the presented protocol does indeed deliver specific isolation of fibroblasts that can be cultured for multiple passages without developing strong characteristics of epithelial and adipocyte-like cell lines.

Alternative to undertaking additional experiments, if transcriptomic/expression data exists in the literature that would provide an indication of the expression of these markers in epithelial/adipocyte cell lines or equivalent, this data should be discussed or referenced and ideally presented in a table providing comparison to the current data.

Thank you for the insightful feedback to improve the impact of the data presented in our manuscript. We took a bioinformatic approach to providing a point of reference for MEaF transcriptomics by collecting metadata and experimental reads from R packages. Data for 3T3-L1 pre- and post-adipocyte differentiation was collected from the package, curatedadipodata, and pre- and post-mesenchymal transition of Human Mammary Epithelial (HMLE) cell data from the emtdata resource from the R package, experimenthub. Further details for our bioinformatic approach are provided in the methods section and Reviewer 1's major comment #3. To provide additional context, we added rows to Table 2 and reformatted Figure 3C for a clearer comparison of MEaF transcriptomics to reference cell lines.

b. Are experiments performed using appropriate methods that will answer the question (or test the hypothesis or support the observations) posed by the authors? Is the right tool used for the job? All method used is appropriate for testing the protocol established in this manuscript.

Thank you for acknowledging that our methodology is appropriate for testing this protocol.

c. Were the data analysed using appropriate statistical tests?

Data were analysed using appropriate statistical tests. However, as detailed below in 2b, images in Figure 2 and Figure 3 were not analysed and therefore not subjected to statistical testing. This should be rectified as suggested below.

Thank you for the flexibility to amend this concern. As described above, we included at least three separate photos from 3 separate experiments for IFA and Incucyte photos. For Figure 2C-E, now 2D-F, we apologize that it was unclear. These were not pictures taken at each passage but instead passaged for the given amount of time and a picture taken at the end time point. The photos in 2D-F are representative of 2 separate experimental flasks of MEaF cultures.

2. Reproducibility

a. Were experiments in each figure performed using adequate number of biological replicates?

Yes, an adequate number of replicates were used for presented data and this information is now included in the figure legends. However, there were some instances where results were referred to but data not presented. Instances of this were in: Discussion lines 182/183 in reference to isolating cells from smaller tissue inputs; Discussion lines 194-196.

In Discussion lines 182/183, no data is presented (unless I missed it) of fibroblasts isolated from ear hole punches despite the assertion that this was performed successfully. Given the ethical and experimental advantages of this approach, at least presenting a few images to complement those in Fig 3A or Fig 2 C-E would be good to show that such an isolation produces cells with similar morphology to the larger ear isolation.

In Discussion lines 194-196, there is no data to complement the statement about isolation of fibroblasts from a 3-week-old mouse and that adipocyte-like cells were still derived from this isolation. This would be a useful supplementary figure.

We appreciate this reviewer and their interest in our experiments for ethical and experiment optimization by comparing MEaF isolations from different ages of donors and various tissue size for isolation. Images to support our claim of successful isolation of cells from younger donors and two ear hole punches are represented in Figure S4. It is important to note that all of the cells in Fig. 4 were grown in minimal media and adipocyte-like cells do not appear.

Language surrounding these claims have been updated throughout the manuscript:

Results:

“Cells with an accumulation of lipid-like structures are indicative of adipocytes and were found in MEArF cultures in enriched media, regardless of the donor’s genotype (Fig. S2A).”

Discussion:

“We validated the use of two mouse ears per digestion as a reliable source of MEArFs. Consistent with previous isolation protocols (Edelman et al., 2017), we successfully isolated MEArFs using just two ear hole punches without filtering out the tissue, as we suggest doing with larger tissue inputs. MEArFs from two hole punches did take a total of 34 days to reach confluence in a T25 flask, compared to both ears from a 3-week-old donor (7 days to confluence) or a 12-week-old donor (12 days to confluence) (Fig. S4). Our protocol may be adapted by growing the initial culture from 2 to 4 hole punches in a 6cm Petri dish to aid confluency, before transitioning to a culture flask. We did not characterize the long-term proliferative capacity or transcriptomic adaptations of the cells isolated from a smaller tissue input. Non-lethal tissue collection offers the advantage of establishing parallel in vitro models from animals undergoing in vivo experimentation, but will require further optimization.”

And

“When grown in minimal media, adipocyte-like cells and structures are undetectable in MEArF cultures (Fig. 2B). Cells with an accumulation of lipid-like structures were found in MEArF cultures in enriched media, regardless of the donor’s genotype (Fig. S2A). Cells imaged without BODIPY stain demonstrated a lack of specific fluorescence, confirming the BODIPY stain co-localized with the adipocyte-looking cells (Fig. S2B).”

b. Is there sufficient raw data to assess the rigor of the analysis?

For transcript related figures (3, 4, Table 2) sufficient raw data is presented, and RNA sequencing data will most likely be available on NCBI (there is a BioProject ID, but the ID does not yet seem to be active).

For imaging related figures (Figure 2, Figure 3A, B) insufficient raw data is presented, which is especially important given there is no quantification of these images. These images should be quantified where possible. Quantification of the percentage of BODIPY positive cells in Fig 2A,B should be performed. A blinded, manual count of adipocyte like cell formation in Fig 2C-E should be undertaken. The mean vimentin intensity per cell should be quantified in Fig 3B. Where there is an absence of image quantification, a panel of images of each condition from multiple biological repeats needs to be presented in a supplementary figure.

Thank you for these suggestions. We have created an “Editor” viewing link for the SRA Database information at this link:

<https://dataview.ncbi.nlm.nih.gov/object/PRJNA1400943?reviewer=2vtkbump110ualo01m4qfa7ah1>

For Fig. 2A and B, 16 photographs were available, so we quantified the number of BODIPY positive cells and present that data as a new Figure 2C. As clarified above, Figs 2C-E, now labeled D-F, are endpoint pictures, so we do not have sufficient photographs taken for quantification. For Fig. 3, we provide additional photos in the main body of Fig 3, and in S3.

c. Does the methods section provide sufficient detail to permit reproducibility?

The methods are excellent, especially the step-by-step protocol of how to isolate the fibroblasts. The tables and supplementary figures to complement the methods are very useful as well.

Thank you for this comment. We wrote the methods very detailed so it would be easy for the next researcher to perform this isolation technique.

3. Completeness

a. Are the author’s conclusions supported by the data?

For the most part, yes. However, as stated in 1a, additional controls or references to transcript data in literature would strengthen the study and the ability of the authors to assert that the isolation protocol produces a relatively pure fibroblast culture.

A minor point, but the statement related to Figure 4A in the figure legend “Samples cluster by passage number with minimal variation” is a bit strong, given that one of the P3 isolates is very

different looking from the other 3 P3 isolates in the PC analysis. It could be useful to highlight that for the most part biological replicates cluster together, but the apparent P3 outlier highlights the need to perform multiple replicates as outliers may have very different cell characteristics.

We want to thank our reviewers for this very fair critique. The outlier P3 sample in the PCA does not support the strength of language used to describe the results as “minimal variation”. We have updated the PCA plot figure legend and corresponding language in the Results with more nuanced language. “Principal component analysis demonstrated clear clustering of samples by passage number with larger variation across passage 3 compared to passage 9 samples (Fig. 4A).” Figure 4A legend: “Samples cluster by passage number with more variability in the early passage cells.”

b. Are there any flaws in the experimental design that invalidate the approach taken by the authors?

There are no experimental design flaws that invalidate the study or approaches taken.

Thank you for this comment. We wanted these methods to be highly reproducible.

c. Are there experiments that have not been performed, but if true would disprove the conclusion? If yes, and if such experiments would be costly or time-consuming to perform, do the authors acknowledge this in a discussion of the limitations?

While I do not think results from the experiments/analysis of transcripts from other cell lines in literature I suggested in 1a would invalidate the study, they would strengthen the authors ability to conclude their isolation method produces a pure primary fibroblast line. I do not think performing the suggested experiments would take an excessive time to perform, however as I suggested, a non-experimental alternative would be to compare to published transcriptomic studies on epithelial and adipocyte-like cell lines.

We appreciate this reviewer’s comments and flexibility to collect transcriptomics data from previously published datasets for baseline comparisons. We used this suggestion to strengthen our argument that MEaRF cultures are primarily fibroblasts. The details of this experiment are outlined in the Figure 3 legend, methods, and Reviewer 2 comment #1.

4. Scholarship

a. Do the authors cite and discuss the merits of relevant data that would argue against their conclusion?

The authors provide balanced citations of publications that do not necessarily support their findings and provide potential explanations for why their results may differ from those published. A table (Table 1) is also presented summarising findings of other published fibroblast isolation protocols. A comment on whether or not there is any myocyte contamination in the current protocol, as was observed for Bravo et al (presented in Table 1) would be useful (and perhaps speculation as to why the authors current protocol does not seem to have myocyte contamination).

Thank you, we are excited to hear that Table 1 was useful to the reviewer. A line was added to the table in the Barnes et al. 2026 protocol notes column, noting that when grown in minimal media, adipocyte and myocyte contamination does not appear in MEaRFs. The excerpt inserted to Table 1 is detailed here:

“No adipocyte or myocyte contamination reported in minimal media, which is nutrient sufficient for fibroblast isolation.”

Our hypothesis for our ability to minimize myocyte contamination is more explicitly stated in the discussion:

“Myocytes are noted contaminants of published fibroblast isolation protocols, and we did not see such contamination in our cells grown in minimal media. The absence of myocyte contamination in our MEaRFs cells may be due to a few reasons. First, we notably remove tissue chunks immediately after mechanical and enzymatic tissue digestion. Second, myocytes take time to divide and differentiate, so our care of fibroblasts may be selective against the growth of these cells. Finally, we confirmed the absence of myoblasts through the absence of marker genes, Myod and Myf5, in MEaRF transcriptomics.”

b. Do the authors cite and discuss the merits of relevant data that would support their conclusion? Supporting literature is discussed, both in text and presented in Table 1.

Thank you.

c. For techniques/methods manuscripts, Do the authors cite and discuss the current state of the field and clearly explain how the method improves the field?

The authors discuss the current state of the field well for the most part. While the authors make statements about how their method improves the field, some of these statements need additional discussion or supporting evidence.

That this protocol is "cost effective" is stated numerous times, while in other instances the authors say "cost-optimized". "Cost-optimized" should be consistently used because as far as I can tell, the only cost reduction over many other protocols is through using less serum, using 10% FBS compared to 15% FBS in some- but not all- other protocols. For example, Ningshih et al also used 10% FBS and standard DMEM but also used only mechanical disruption of tissue- this would surely be more cost effective than the authors current protocol which includes an enzymatic digestion step, with the added cost of the enzyme. I assume enzymatic digestion helps with the yield and purity of the isolation, hence it being a "cost-optimized" protocol. The authors could provide comment on why the liberase digestion aids the protocol.

The authors also state their protocol prevents non-fibroblast contamination, which it clearly does regarding adipocytes. However, given other protocols they have cited found myocyte contamination of their fibroblast cultures, they authors should comment on the potential for such contamination in their protocol, and why their protocol avoids this contamination. Providing more phase contrast images as in Figure 2A and B in a supplementary figure to show there is minimal myocyte contamination across a large number of cells/ fields of view would provide a useful addition to the paper as well.

We agree, this protocol is cost-optimized, rather than entirely cost-effective. Although not a primary focus of this manuscript, this fibroblast isolation protocol can be further optimized by identifying the minimal amount of FBS and the cheapest tissue digestion enzyme.

It is important to point out the limitation of this study that the experiments were only done in the outlined conditions of this protocol, and we can only speculate on the downstream behavior of MEArFs with a different tissue digestion reagent or media components.

We have added a comment on our decision to use the Liberase enzyme as mentioned above in Reviewer 1 Major Comment #6.

Additional fields of view are provided throughout the new version of this manuscript, as highlighted above and exhibited in Figures 2A, 2B, 3A, S4, including BODIPY unstained controls in Fig. S2A, secondary-only IFA photos in Fig. S3.

Adjustments to the photos brightness and contrast are mentioned above in Reviewer 1 Major Comment #1, and detailed in figure legends, and methods.

Myocytes were not detected in the additional planes across three separate isolations in Fig. S4, or in the transcriptomic analysis of passage 3 and passage 9 MEArFs as discussed in Reviewer 2's comment #4 on Scholarship.

Second decision letter

MS ID#: bio.062483R1

MS Title: Isolation, Validation, and Long-Term Culture of Mouse Ear Fibroblasts

Authors: Katie L. Barnes, Nicole M. Davis, Billy J. Erazo, Katie M. Cataldo, Emmaly H. Bertges and Laura Knoll

I am happy to tell you that your manuscript has been accepted for publication in Biology Open, pending our standard publication integrity checks. It was accepted on 17th April 2026.